# Post-wildfire sediment source and transport modeling, empirical observations, and applied mitigation: an Arizona USA case study

**Edward R. Schenk[1], Alex Wood[2], Allen Haden[2], Gabriel Baca[2], Jake Fleishman[2], and Joe Loverich[3]**

[1]Stormwater Section, City of Flagstaff, 2323 N. Walgreens Dr. Flagstaff, Arizona USA 86004,
[2]Natural Channel Design Engineering, Inc., Flagstaff, Arizona USA,
[3]JE Fuller Hydrology and Geomorphology Inc., Flagstaff, Arizona USA

*Correspondence to:* Edward R. Schenk, edward.schenk@flagstaffaz.gov

## Abstract

Post-wildfire floods are receiving greater attention as wildland-urban interfaces become more common and as catastrophic wildfires have increased in frequency. Sediment sourcing, transport, and deposition in the post-wildfire environment receive attention due to the severity of risk caused by debris flows and concentrated sediment flood flows. This study compares sediment model predictions based on MUSLE, the WARSSS suite of models, and the free internet-based WEPPcloud post-wildfire sediment modelfor the 2019 Museum Fire (809 Ha of steep slope *Pinus ponderosa* forest on a series of basaltic domes). Empirical evidence from four floods in 2021 indicated 9,900 Mg of sediment yield to city of Flagstaff neighborhoods, WEPPcloud estimated 3870 Mg/year, MUSLE predicted 4860 Mg/year (based on the four events), and the WARSSS suite of models predicted 4630 Mg/year. Both WEPP and WARSSS estimated more sediment yield from channels than hillslope (51%/49% and 60%/40% respectively) though the spatial patterns differ between the models. The utility of sediment forecasting to inform the application of sediment mitigation structuresto help reduce downstream impacts of post-wildfire water and sediment flows is discussed. Continued revisions of sediment forecasts, based on case studies such as this one, can provide researchers, managers, and policy makers with tools for ecological and human risk mitigation and emergency management.

## 1. Introduction

Post-wildfire flooding at the wildland-urban interface (WUI) is an increasingly important issue for the health and safety of millions of humans living in or adjacent to semi-arid forests (e.g. Ebel et al. 2023; Kinoshita et al. 2016; Sankey et al. 2017).  The development of neighborhoods directly adjacent to forest lands under severe drought conditions creates hazards not just to widespread burning but flooding in the aftermath of those fires (Kinoshita et al. 2016; Sankey et al. 2024).  While the changes in hydrologic properties of watersheds after severe wildfires are relatively well known, there is now a need to rapidly assess and mitigate post-wildfire sediment transport and floods to prevent, or lessen, impacts to safety and property damage. A key portion of this process is understanding the potential for damaging debris flows and sediment sourcing, transport, and aggradation (Moody et al. 2013; Smith et al. 2011). The sediment component of the post-wildfire flood paradigm is perhaps the most damaging due to physical impacts (e.g. damage to infrastructure), bulking factors to flood flow volume, and long-term damage to soil profiles and stream channels that hamper watershed ecosystem recovery (e.g. Moody et al. 2013; Neary et al. 2011; Shakesby 2011).

In the past two decades there has been an emphasis on predicting, and remediating, post-wildfire sediment sourcing and transport (Shakesby et al. 2016). Most studies have been focused on hillslope and channel process and determining accurate, or precise, estimates of sediment fluxes (East et al. 2021; Rengers et al. 2021; Wu et al. 2021). While many studies rely on empirical measurements there have been recent improvements and updates to post-wildfire sediment modeling. These improvements allow

for rapid, relatively inexpensive, assessments post-wildfire but potentially at the cost of precision or accuracy (Lopes et al. 2021).

This study explains sediment prediction methods utilized for a relatively small wildfire in Flagstaff, Arizona USA to predict sediment quantities as well as flow paths and sedimentation areas for the
Museum Fire (2019). These methods have successfully guided mitigation efforts for the nearby Schultz Fire (2010; Neary et al. 2012) and were also used in another nearby fire in 2022 (Pipeline Fire). Three models were compared to empirical observations to provide an estimate of model precision and accuracy, the Watershed Assessment of River Stability and Sediment Analysis (WARSSS), Modified Universal Soil Loss Equation (MUSLE), and the Watershed Erosion Prediction Project (WEPP), all three
were then compared to field observations immediately downstream of the modeling domain in the urban environment.

Additionally, sediment mitigation structures (alluvial fan restoration areas) are discussed at the end of this study and are described as "work areas". These work areas include alluvial fan restorations that spread flow allowing for a drop in stream power. The loss in stream power allows for sediment accretion
upstream of urban neighborhoods (Grover 2021; Rosgen and Rosgen 2015). The work areas were informed, and designed, by the 2022 modeling effort and were built in late 2022 and early 2023 after the 2021 floods.

The objective of this study is to provide an estimate of the precision and accuracy of three sediment modeling techniques (WARSSS, MUSLE, and WEPP) and determine where future modeling
improvements should be focused upon. All three modeling techniques are available internationally and with low barriers to entry, either extremely user friendly (WEPP in the WEPPcloud user interface) or with large applied technical user bases (MUSLE and WARSSS). A secondary objective is to display the utility of post-wildfire sediment modeling for determining the location, and type, of sediment mitigation structures. The intent of both objectives is to help researchers and land managers rapidly, and accurately,
predict post-wildfire sediment risk to both ecological and human environments so that sediment mitigation strategies can be developed and implemented to reduce risk to life and the ecosystem.

## 2. Study Site

Flagstaff, Arizona lies at the edge of the dormant San Francisco Volcanic Field including the San Francisco Peaks, Dry Lake Hills, and Mount Elden. The local watersheds are generally hydrologically
complacent, unless disturbed, with extremely low rainfall-runoff ratios due to local geology (weathered dacite, cinders, and karstic fractured limestone), vegetation (dense *Pinus ponderosa* forest), and relatively deep soil organic layers (Quisenberry 2009; Youberg et al. 2019; Schenk et al. 2021). The Spruce Wash watershed is an ephemeral tributary to the Rio de Flag, another ephemeral watershed that drains the southern portions of the San Francisco Volcanic Field. The Spruce Wash watershed drains the
six dacite intrusive hills that make up the Dry Lake Hills feature (2695 m) as well as the western portion of Mount Elden (2835 m), a larger protuberance of the same orogeny (Holm 2019; Schenk et al. 2021). A previous USGS study observed a peak flow of 0.14 cubic meters per second in the Spruce Wash watershed over a period of 11 years (Hill et al. 1988) despite a watershed contributing area of greater than 1450 hectares.

The Museum Fire occurred in July 2019 over 800 hectares on the steep, mountainous slopes of Dry Lake Hills and Mount Elden, both of which are immediately uphill of established residential areas of Coconino County (CC) and City of Flagstaff (CoF; Figure 1). Mount Elden Estates (MEE; 2160 m) is a rural residential area and is the uppermost residential area within the Spruce Wash Watershed. Approximately one and half kilometers downstream and separated by open U.S. Forest Service (USFS) land are the
urban residential areas of Paradise/Sunnyside (2120 m), which are within the CoF city limits. MEE is located on flatter slopes near the base of Dry Lake Hills on the leading and lower edge of a previously inactive alluvial fan (activated post-wildfire, previously complacent; Fulé et al. 2023). Paradise/Sunnyside are on the toe of inactive alluvial fans and adjacent to the broad, ephemeral, and formerly un-channelized Spruce Wash. Prior to the Museum Fire, the Paradise/Sunnyside neighborhoods
had one defined channel/pipe system and surface water flow seldom occurred within these existing channels. Up gradient on USFS land, ephemeral surface flows were spread over wide alluvial fans (areas of sediment deposition) and were easily absorbed into the unconsolidated sediment. Consequently, pre-

wildfire surface water flows within the channels were primarily from stormwater runoff during normal precipitation events from local CoF streets (Schenk et al. 2021; Schiefer and Schenk, 2024).

## 105    2.1 Flood Events and Study Timeline

The Flagstaff region saw record low summer monsoonal rain in 2019 and 2020 with no substantial post-wildfire impacts. Initial post-wildfire induced flooding occurred during the above average summer monsoon season of 2021, resulting in several debris flows high within the Spruce Wash watershed and four significant floods that entered the downstream city (Porter et al. 2021; Porter et al. 2023; Schenk et
al. 2023; Sankey et al. 2024). Post-wildfire flooding resulted in vast amounts of sedimentation in downstream residential areas as existing drainage features and channels were overwhelmed with sediment and debris (e.g. https://www.weather.gov/fgz/FlagstaffJuly2021). These 2021 flood events allowed for empirical observation of sediment discharge and eventual comparison to the modeled predictions of sediment discharge. The models (WARSSS, MUSLE, and WEPP) were run in late 2021
and early 2022 after the flood events, model results were directly comparable to the 2021 flood observations and also used to inform the location and design of the alluvial fan restoration sediment mitigation structures, which were built in late 2022 and early 2023.

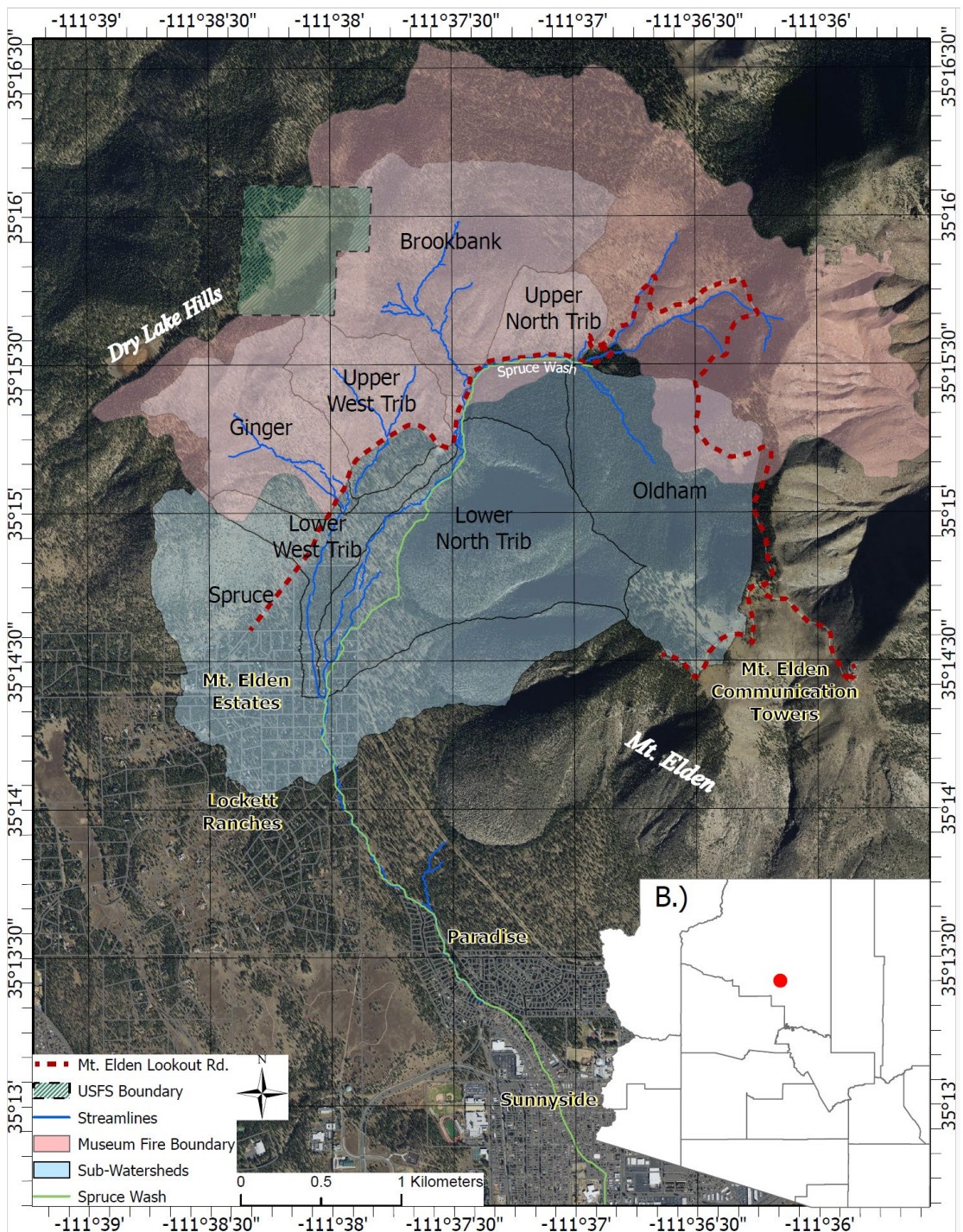

**Figure 1. Overview map of the 2019 Museum Fire watershed (Spruce Wash), sub-tributary names, and locations of impacted neighborhoods. The City of Flagstaff is located in north-central Arizona on the edge of the Colorado Plateau and shown on the lower portion of this figure. Background aerial imagery is public domain (Coconino County 2015).**

## 3. Methods

## 3.1 Flood Flow Modeling

Post-wildfire flood modeling was completed in August 2019 and was based on a 2-D hydrologic-hydraulic numerical model created in FLO-2D (JE Fuller 2019; JE Fuller 2022). Initial flood modeling was completed at a 20-foot (6m) grid scale using 2015 lidar elevation data, subsequent modeling was

completed at a 5 foot (1.8m) grid scale using a fall 2019 lidar elevation dataset, both datasets are
available on the United States Geological Survey National Map portal
(https://www.usgs.gov/programs/national-geospatial-program/national-map). All modeling indicates an
approximate 10 to 100 times (one to two orders of magnitude) increase in surface water peak discharge
depending on rain event; more information on hydrologic conditions is provided in a conference
proceedings paper (Schenk et al. 2023).

## 3.2 Sediment Modeling (WARSSS and MUSLE)

Sediment modeling focused on quantifying relative sediment sources relating to channel and hillslope
erosional processes. The Spruce Wash watershed within the Museum Fire burn scar was divided into
sub-watersheds, based on tributaries in the USGS National Hydrography Dataset, to identify high-
sediment yield areas (USGS 2019). Low gradient areas downstream from high sediment yield areas were
identified as "work areas" for applied sediment control practices that have the greatest impact on limiting
downstream sediment transport (colloquially described as "alluvial fan restorations" elsewhere;
https://www.coconino.az.gov/2407/Alluvial-Fan-Stabilization-Project; Rosgen and Rosgen 2015).

The WARSSS model (Rosgen 2009), was the first modeling suite used for this fire (in 2021 post-
flooding) due to its successful sediment transport predictions after the nearby 2010 Schultz Fire (NCD
2012, Neary et al. 2012). WARSSS is designed to identify the location, nature, extent, and consequences
of land use impacts on sediment and understand the cause of watershed impairment. This approach was
developed for application on large watersheds and is practical for the Museum Fire because it uses
previously proven, rapid screening field observations that integrate hillslope, hydrologic, and channel
processes.  The analysis focuses on average annual yield of sediment rather than event-based analyses.
The average annual yields do not ignore sediment delivery from large flood events but take into account
the overall frequency of these types of flows, based on a 30 year climate average.  An annual average
sediment yield is used due to the highly heterogenous precipitation distribution in the American
Southwest during monsoon storms. This annual average sediment yield, therefore, is appropriate for
understanding watershed function and developing watershed restoration practices post-disturbance.

The WARSSS method relies on estimating bank erosion using the Bank Assessment of Non-Point
Source Consequences of Sediment (BANCS) model and can quantify bank erosion rates and sediment
supply for years with normal discharge patterns (Rosgen 2009). Average annual hillslope erosion is
estimated using the Erosion Risk Management Tool (ERMiT; Robichaud et al. 2014). The Modified
Universal Soil Loss Equation (MUSLE; Williams and Benhardt 1977) is utilized to estimate sediment
supply from hillslopes during specific precipitation events. Discharges for these events were estimated
by a 2-D numerical model with NRCS curve number inputs as part of the post-wildfire flood modeling
efforts (JE Fuller 2019; Schenk et al. 2023).  The MUSLE estimates are provided here as a reference
point for larger events. Direct comparison of the different methods is difficult.  While post-wildfire
hillslope erosion will diminish over time with natural recovery, sediment bank contributions are expected
to continue at high rates for many years due to post-wildfire channel evolution processes, from initial
incision immediately post-wildfire (Santi et al. 2008) to subsequent widening of channels through time
(Benda et al. 2003; Hupp and Simon 1991).

Sediment transport estimates were used to look at how supplied sediment can transport through the
channel system. Sediment transport modeling used the FLOWSED/POWERSED platform in the
RiverMorph software (v. 5.1) and provided estimates of average annual sediment transport through a
specific cross section of channel given an annual flow scenario (Rosgen 2009; Hall and Bledsoe 2023).
Estimates of sediment supply into a reach can be compared within the reach to aggradation or
degradation for both existing and proposed design. This analysis is sensitive to several data inputs
including annual flow duration curves (based on watershed size), bankfull discharge, suspended
sediment and bedload sediment rating curves, channel configuration and slope (Rosgen 2006; Hall and
Bledsoe 2023). These data are difficult to obtain for ungauged ephemeral systems; we used sediment
rating curves and dimensionless flow duration curves developed during the 2010 Schultz Fire sediment
analysis which were derived from regional data and research from the Beaver Creek Research watershed
effort (Natural Channel Design 2012).

Once high-sediment yield areas are identified, sediment transport analyses are conducted at typical
channel cross-sections that typify the range of channel conditions from upstream to downstream,

proposed work areas in the Spruce Wash watershed. In addition to providing an analysis of sediment transport across channels in their current state (fall 2021), an analysis of sediment transport across a conceptualized design channel (a hypothetical 2% slope post-restoration) was used to understand the feasibility of altering the downstream sediment delivery and was based on the upstream sediment supply.

3.3 Assessing the geomorphic channel condition
       3.3.1. *BEHI data collection*- Bank erosion hazard index (BEHI) surveys were used to qualitatively evaluate all eroding channels within the Spruce Wash watershed, (Rosgen 2009; Figure 2). Collected data consisted of channel bank height, channel and bank material, length of channel, vegetation and root density, bank slope angle, valley and stream type classification (Rosgen 1996), and near bank
stress (NBS).

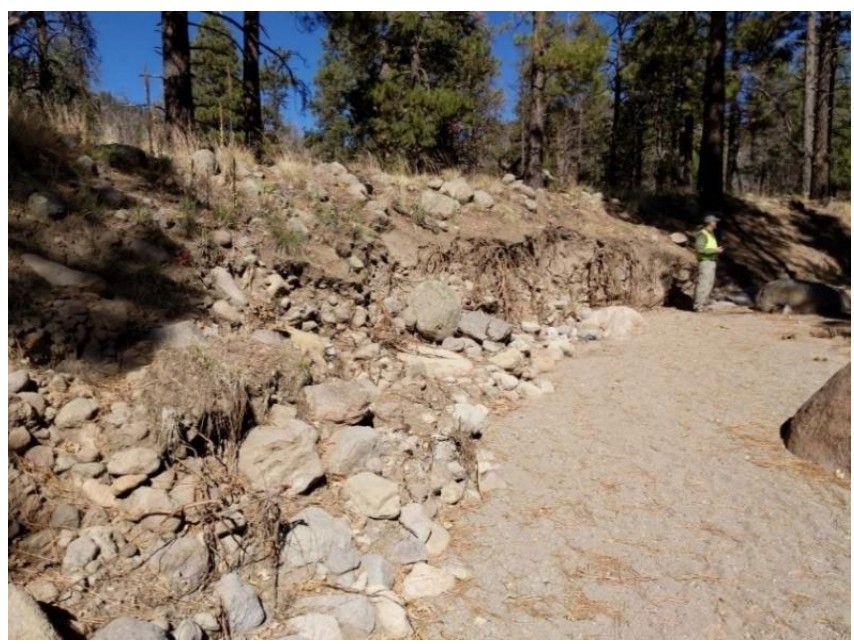

**Figure 2. Surveyors performing a BEHI analysis on a typical channel along Spruce Wash. Photo taken in fall 2021 approximately 500 m upstream of Mount Elden Estates.**

3.3.2. *Channel Surveys*- channel cross-sectional surveys were completed proximal to proposed work areas (i.e. flood mitigation capital improvements) to accurately model sediment transport through channels and assess channel characteristics. Twenty seven (27) cross sectional surveys (Figure 3), longitudinal channel profiles, and pebble counts were completed to evaluate the channel slope and characteristics of specific channel reaches.

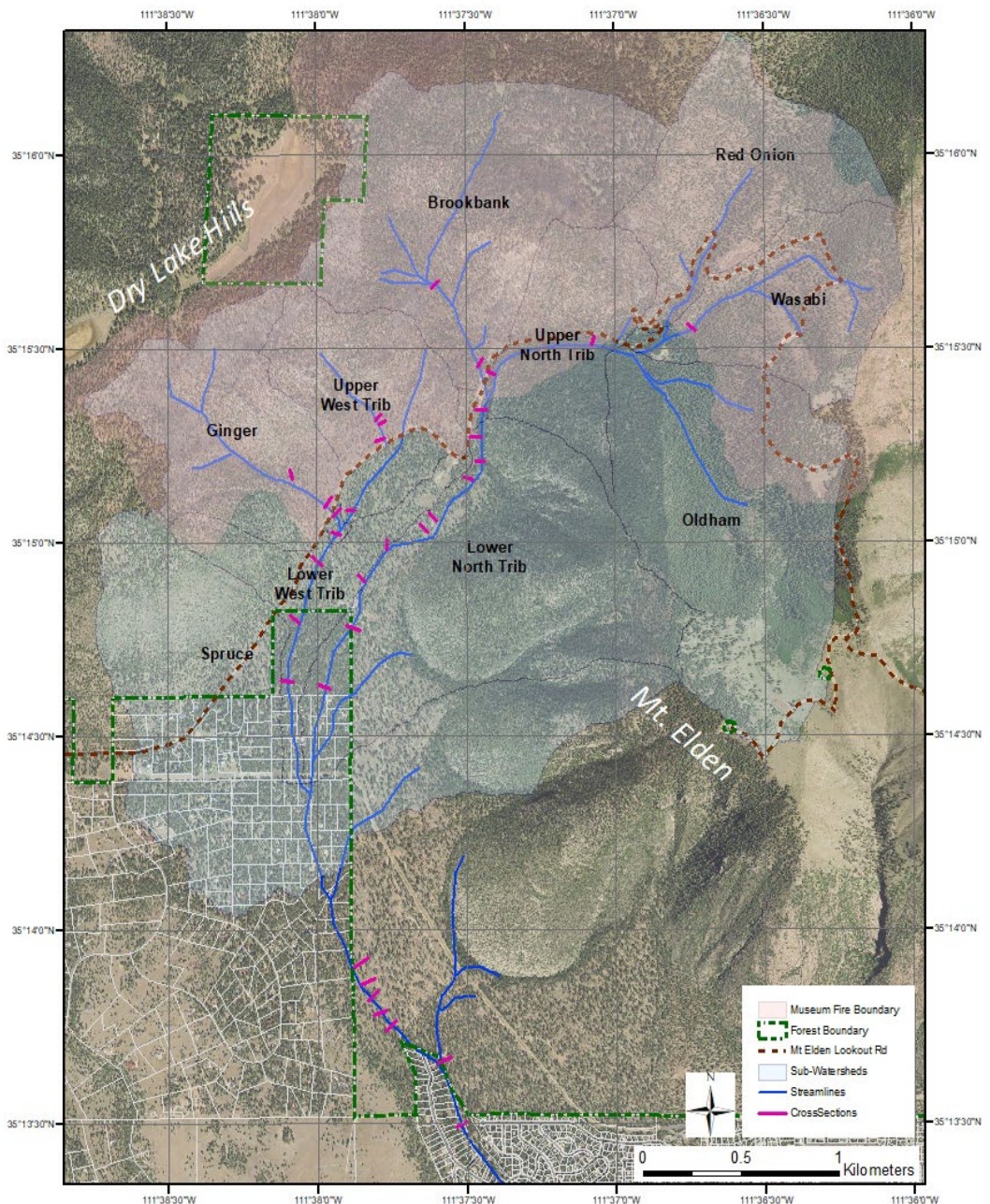


**Figure 3. Cross sections within the Spruce Wash watershed, the 2019 Museum Fire burn area is shown in light red. Background aerial imagery is public domain (Coconino County 2015).**

3.4 Estimating Sediment Yield

3.4.1 Channel Sediment Yield- The Bank Assessment for Non-point source Consequences of Sediment (BANCS) model was used to estimate annual sediment yield (Rosgen 2009). The BANCS model utilizes BEHI and NBS survey data to estimate sediment supply from channel bank sources and yields a sediment supply in mass per year. The BANCS model provides reliable estimates of bank erosion but can underestimate bank erosion rates resulting from higher-than-normal flooding and overestimate

rates from years with very low peak flows (Rosgen 1996). Channel sediment supply was converted to Mg/year/longitudinal meter for all evaluated reaches.

3.4.2 Hillslope Sediment Yield- The Erosion Risk Management Tool (ERMiT) and Modified Universal Soil Loss Equation (MUSLE) model were used to estimate hillslope sediment yield. The ERMiT model predicts sediment yield annually while the MUSLE model is based on precipitation events

(storm-based).

The ERMiT models uses soil burn severity, vegetation type, rock content, hillslope gradient, soil type, hillslope length, and annual precipitation to model sediment yield (Mg/year) up to five years post fire (Robichaud et al. 2007). For the scope of this analysis, 2021 was used as the second-year post-wildfire. Therefore, only years 3 (2022), 4 (2023), and 5 (2024) sediment yield were modeled. To capture the

variability in hillslope impacts, the Spruce Wash watershed was subdivided into sub-catchments using watershed delineation in ESRI ArcMap 10.8 (ESRI 2020). Each catchment was evaluated individually for its sediment yield.

The MUSLE is based on the Universal Soil Loss Equation (USLE) but utilizes transport efficiency and soil erodibility (Igwe et al. 2017). For the post-wildfire watersheds, the MUSLE model is useful for

modeling post-wildfire sediment yield because soil erodibility increases due to hydrophobic, ash laden soils, and transport efficiency increases due to increased runoff from decreased infiltration and retention. The MUSLE model input for post-wildfire situations requires instantaneous peak discharge and total volume of one, two, and three inches (2.5, 5, and 7.5 cm respectively) precipitation in one hour events in addition to watershed area, slope, and soil erodibility. Unlike the ERMiT model, the

MUSLE model predicts event-based sediment yield in mass per event. Soil erodibility (K values) were estimated for low, medium, and high erodibility at 0.29, 0.545, and 0.8 respectively based on field conditions. The crop factor (C value) was estimated at 0.003 for forested area and the slope type (P factor) was inputted as 1 to indicate steep slope. Since the P factor does not provide a measure of the slope the LS coefficient (slope length) was set at 0.5 to account for steep slopes.

3.5 Observed Sediment Transport and Aggradation

Observed sediment transport and aggradation were collected from CoF staff during 2021 flood events (three in July and one in August). Sediment was assessed qualitatively using photographs of known cross sections as well as quantitatively through landfill tipping fees for sediment removed from the channel and streets post-event. Landfill tipping fees were used as a surrogate for sediment deposition

mass, as the landfill calculates fees based on precision scale measurements of truck loads. Each truck load of flood related sediment was measured for potential Federal and State disaster reimbursement, providing a relatively accurate empirical measure of sediment flux to the ultimate outfall (city of Flagstaff neighborhoods).

3.6 Evaluating Sediment Transport and Retention

FLOWSED/POWERSED, a part of the proprietary RiverMorph software package (Rosgen 2006), was used to model sediment transport through channels in their current condition (2021) and through conceptual redesigned channels for mitigating sediment transport. FLOWSED/POWERSED predicts average annual sediment transport (Mg/yr) at a stream reach scale based on flow duration curves for the reach, sediment rating curves for discharge, and the stream power at each stage as determined by

channel morphology. Modeling is based on a typical channel cross section at a riffle within each reach. Stream power is calculated at stage intervals depending on cross section shape. Flow duration at each stage and sediment transport rates are utilized to estimate total sediment load at each stream stage then summed for a total average annual sediment capacity at each reach.

For this study, dimensionless flow duration curves were derived from data collected at nearby watershed

studies in Beaver Creek watersheds (Baker 1982). The dimensionless curves were adjusted using the new estimated 'bankfull' discharge which was derived from runoff models developed for the Museum Fire area (Schenk et al. 2023). The one year return interval precipitation event was used for "bankfull" discharge and approximates the post-wildfire channel forming discharge. Dimensionless sediment rating curves for the project area were derived from sediment rating curves for non-equilibrium condition

watersheds in the Beaver Creek watershed studies (suspended sediment) and non-equilibrium condition post-wildfire watersheds in Colorado, USA (Rosgen 2010; Rosgen and Rosgen 2015).

Based on preliminary sediment yield analyses, FLOWSED/POWERSED was modeled at eight proposed work areas. Each analysis consisted of an existing upstream sediment source cross-section and a proposed alluvial fan restoration cross-section. Upstream sediment source geometries were obtained from previously completed geomorphic surveys. Each analysis was iterated using the same upstream sediment source cross-section and a conceptual (proposed) design cross-section. The design cross-section informed the final work area cross-section and was drawn in RiverMorph to incorporate a design that promotes the greatest amount of sediment retention.

For each model run, FLOWSED and POWERSED required the following inputs: bankfull cross-sectional area, Manning's n value, bankfull discharge, slope, suspended sediment (mg/L), measured bankfull bedload (lb/s), a flow duration curve, and a sediment rating curve comparison (data available in: NCD 2022 under Appendix B). FLOWSED models the total annual sediment yield, both suspended and bedload, using flow-duration curves and their corresponding sediment yields. The dimensionless flow-duration curve is developed from representative watersheds in the region using USGS stream gage data. The POWERSED model compares sediment transport in various configurations of channel geometry.

The FLOWSED/POWERSED model was used to estimate the effect of rebuilding alluvial fans to increase sediment retention upstream of the city. A conceptual design cross-section was used at each work area and evaluated for its efficiency in sediment transport. Design cross sections consist of a restored fan feature with the eroded, defined flow paths graded flat and stabilized with lateral rock sills. This added sediment retention was accomplished by widening and repairing the existing channel into a designed alluvial fan channel to fill the valley bottom. This reduces the ability of the channel to transport sediment by lowering shear stress and stream power. The slope of the channel remains the same, but the depth is lowered by allowing for a wider flow path.

### 3.7 Internet-based Sediment Source and Transport Modeling (WEPPcloud)

WEPP (Watershed Erosion Prediction Project) model runs were completed using the WEPPcloud online toolkit (http://wepp.cloud/weppcloud/) in 2022 to compare with the WARSSS suite of models presented above. WEPP is a standard post-wildfire sediment tool for the US Forest Service and has been expanded in the last decade to include an online modeling tool based on available topography, soils, and climate data for three continents (Lew et al. 2022). The modeling domain is largely based on the Soil and Water Assessment Toolkit (SWAT) methodology with adjustments based on empirical relations since the initial SWAT development (Dobre et al. 2022). The post-wildfire "disturbed" WEPP model was populated using the USFS BAER team soil burn severity georeferenced raster file for the Museum Fire (available through the USFS Inciweb portal; https://inciweb.nwcg.gov/) and model runs were completed using the Cligen precipitation toolbox with a PRISM modified climate application (see Dobre et al. 2022 and Lew et al. 2022 for more information). The model outlet downstream condition was selected at the Spruce Wash entry into the CoF neighborhoods (Linda Vista Avenue; 35 13'22.74", 111 37'31.03").

## 4. Results

### 4.1 Channel Conditions

Approximately 20% of the channels in the Museum Fire watershed are incised channels with high sediment contribution from channel and bank processes ("G" type channels in the Rosgen classification). G channels were found primarily in the burned, steep, upper reaches of the watershed; however, some were found in reactivated alluvial fans (Figure 4). Bank erosion from this type of channel can be an order of magnitude higher sediment contribution from bank and channel processes than other non-incised steep slope channels (Rosgen 2009). Aggrading, often braided, "D" type channels or valleys that can support aggrading alluvial fans, or riparian floodplains, are roughly 15% of the watershed. While these channels have the potential to store large amounts of sediment, many are gullied post-wildfire and now function as sediment sources rather than sediment sinks. The Rosgen channel type was determined visually during the BEHI surveys, and the results are shown in Figure 4.

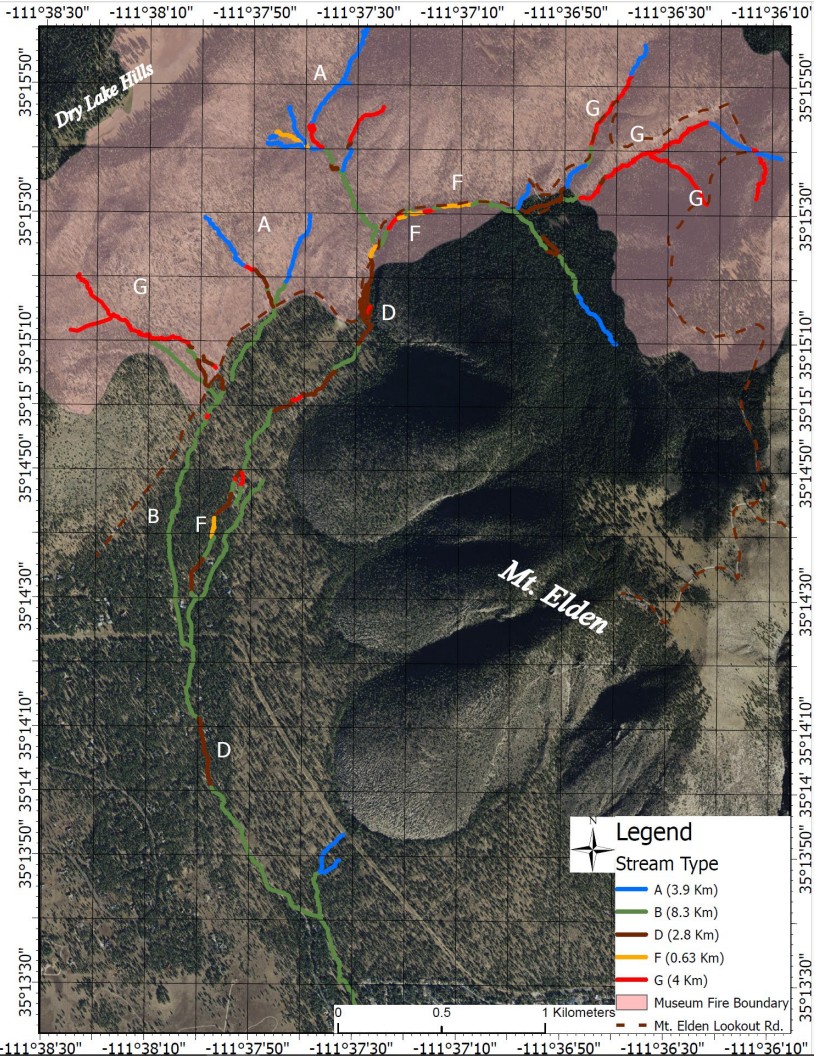

**Figure 4. Spruce Wash channel types based on the Rosgen classification system. "A" and "B" types are generally stable with low sediment contribution. "F" and "G" channel types are generally unstable and are sediment sources, "D" channel types tend to be aggradational (Rosgen 2009). Background aerial imagery is public domain (Coconino County 2015).**

### 4.2 Sediment Yield

4.2.1 *Channel and Hillslope (ERMiT) Sediment Yield*: The BANCS model estimates a total sediment yield of 9,408 Mg/yr from streambank erosion while the ERMiT model estimates that hillslope erosion would yield 6,300 Mg of sediment in 2022. Combining both methods, sediment yield resulted in a cumulative 15,720 Mg/yr of predicted sediment yield from channels and hillslopes in their current conditions for the year 2022 (3 years post-wildfire; Table 1 and Figure 5). However, these channels do not have the capacity to transport the entire sediment source to the city, the POWERSED/FLOWSED models (Figure 5 and 6) take transport capacity into account and indicate a transport rate of 4630 Mg/year on average. Empirical observations by CoF staff were 9,900 Mg of sediment delivered to the downstream end of the study site in 2021 from four flood events, the majority of the sediment transported to the city was during the first flood event, despite the magnitude of the flood event being less than some subsequent floods (Schenk et al. 2023).

The BANCS model also estimates the unit bank erosion rate which is the erosion rate per longitudinal length of channel (0.3m in this case). Figure 7 presents the unit bank erosion rate for channels in the Spruce Wash watershed, indicating the channels with the highest expected erosion rates. The Ginger and

Wasabi sub-watersheds, which are two steep watersheds in the burn area, have the highest unit bank erosion rates. The results of the ERMiT model showing the predicted hillslope erosion rates are presented in Figure 8 which generally show the highest hillslope erosion rates in the steeper, burned areas of the watershed.


**Table 1. BANCS, ERMiT, and total predicted sediment yield for Spruce Wash sub-watersheds. BANCS modeled bank erosion is a result of a channel survey of current condition while hillslope erosion is determined as a year 3 post-wildfire ERMiT modeled sediment yield. Bold numbers indicate sub-watersheds where hillslope erosion is predicted to be larger than bank erosion. Values are provided as shown in the model output, precision is likely to the hundredths place. Total predicted sediment transport (as modeled by POWERSED/FLOWSED) is shown at the bottom as well as empirical field observations from 2021 flood events.**


| Sub-Watershed | Bank Erosion | Hillslope Erosion in 2022 | Total Erosion | Area | Total Unit Erosion |
|---|---|---|---|---|---|
| | (Mg/year) | (Mg/year) | (Mg/year) | (Hectares) | (Mg/year/hectare) |
| Brookbank | 1006 | **1986** | 2,992 | 163 | 18 |
| Ginger | 2670 | 1152 | 3,822 | 87 | 44 |
| Lower North Tributary | 654 | 3 | 657 | 170 | 4 |
| Lower West Tributary | 327 | 1 | 328 | 51 | 7 |
| Oldham | 317 | 242 | 559 | 163 | 3 |
| Red Onion | 536 | **985** | 1,521 | 90 | 17 |
| Spruce | 363 | 4 | 367 | 210 | 2 |
| Upper North Tributary | 319 | 303 | 622 | 62 | 10 |
| Upper West Tributary | 460 | **606** | 1066 | 69 | 15 |
| Wasabi | 2757 | 1028 | 3785 | 74 | 51 |
| **TOTAL Source Sediment** | **9,408** | **6,309** | **15,717** | | |
| **TOTAL Sediment Transport** | | | **4,630** | | |
| **TOTAL Observed** | | | **9,900** | | |

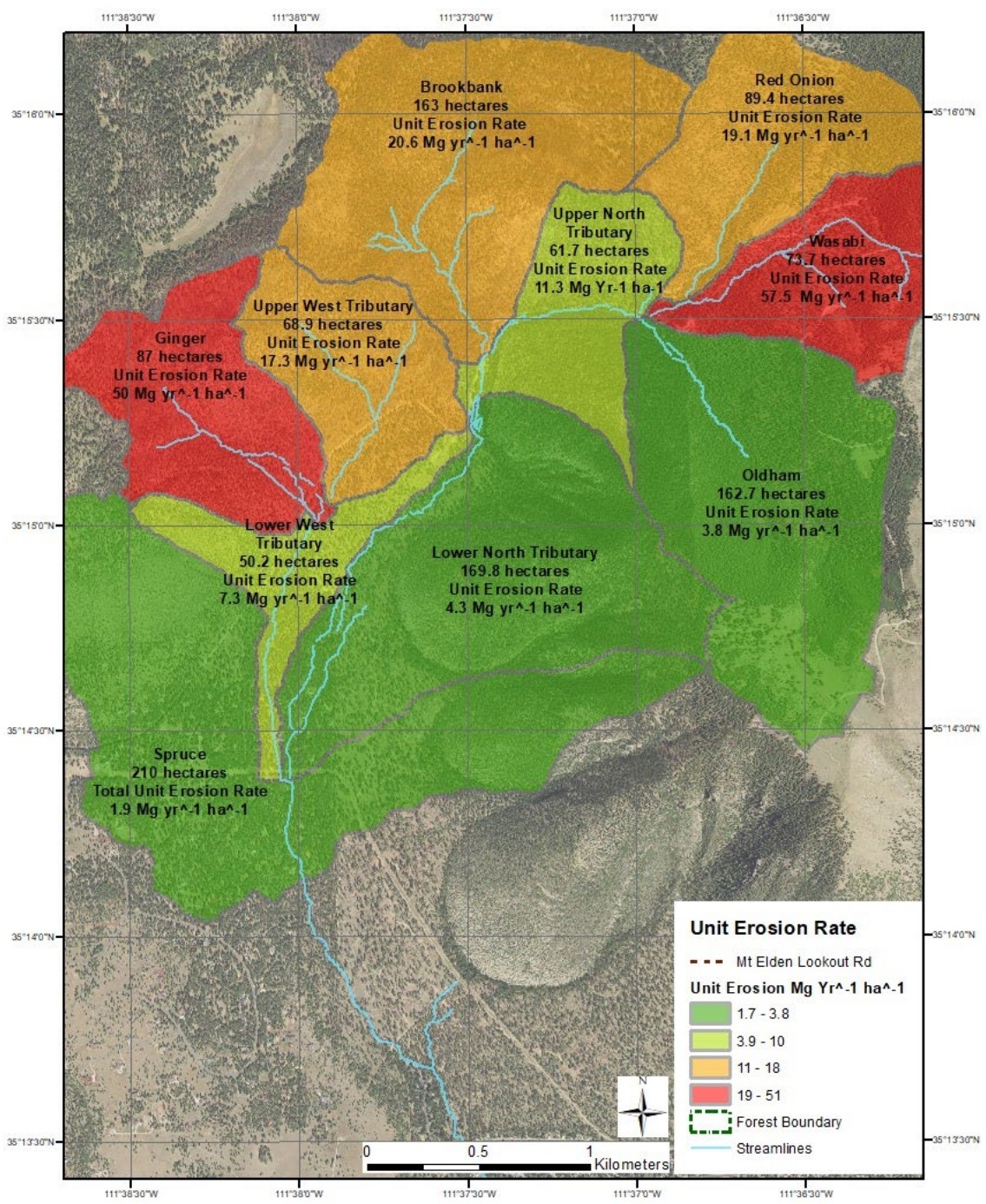

**Figure 5. Unit erosion rates for each sub-watershed based on the ERMiT model. Background aerial imagery is public domain (Coconino County 2015).**


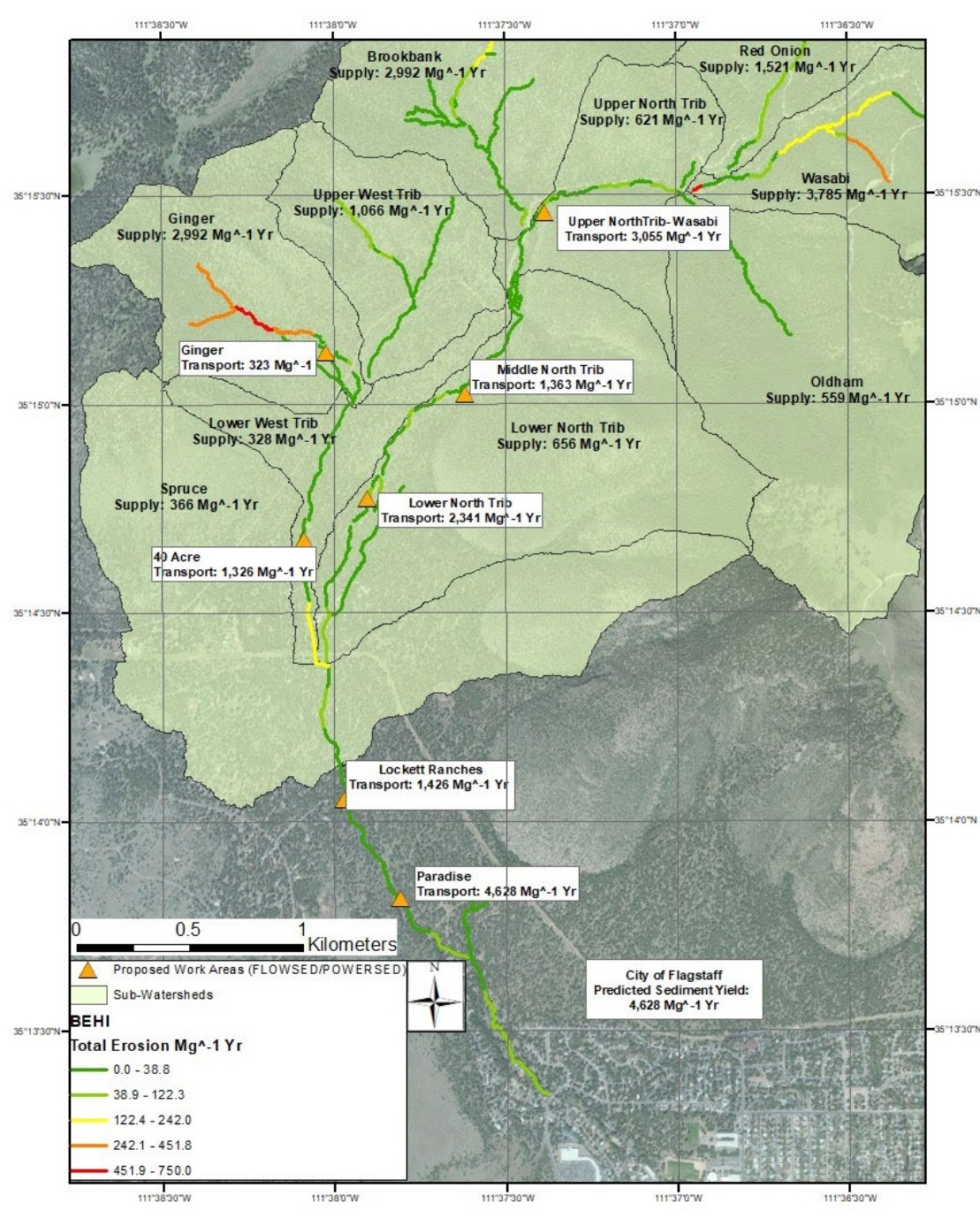

**Figure 6. Sediment transport capacity for all work areas (constructed in 2022 and 2023) from the POWERSED/FLOWSED model. Background aerial imagery is public domain (Coconino County 2015).**



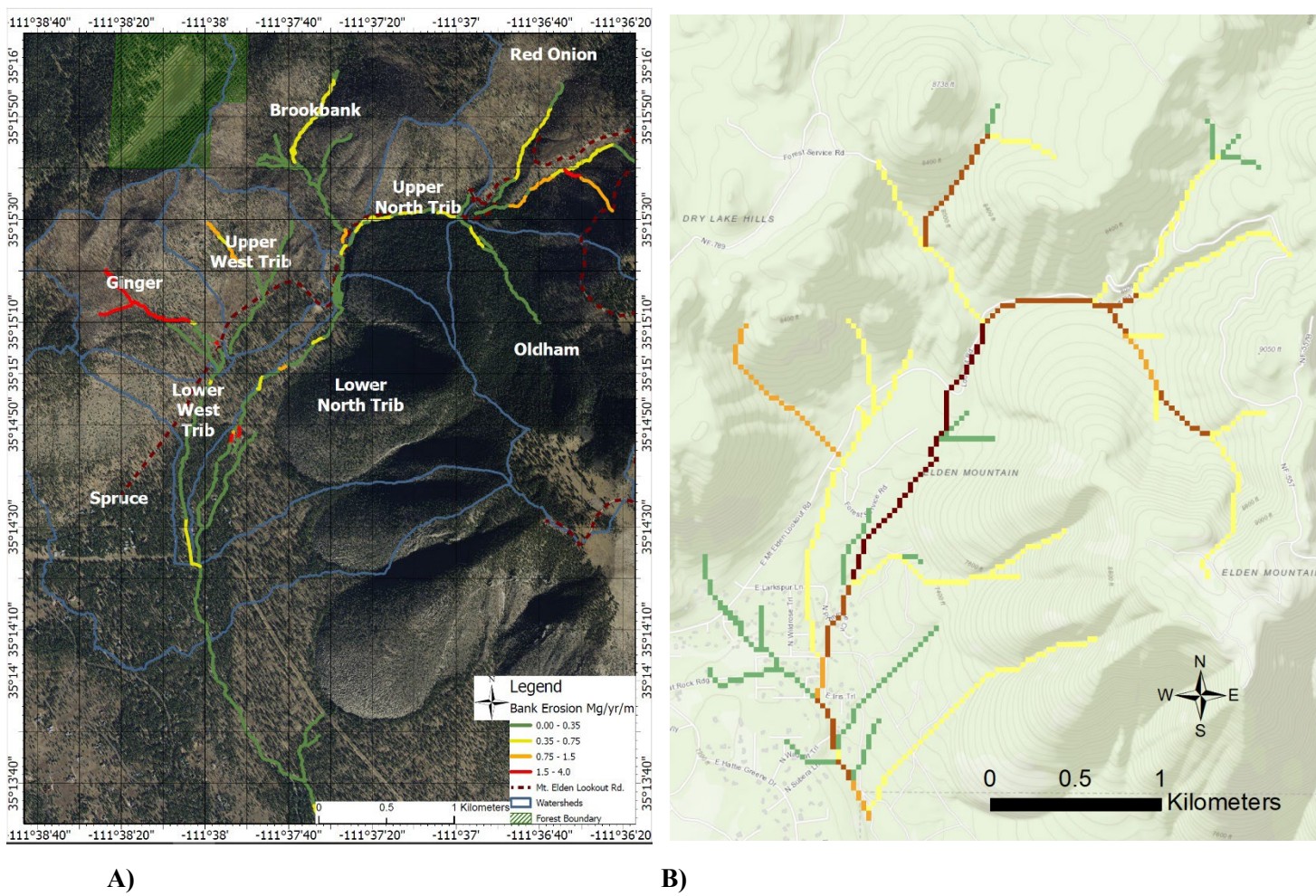

A)                                                                                      B)

**Figure 7. A) BANCS modeled bank erosion rates for tributaries within the Spruce Wash watershed. B) WEPP modeled bank erosion, there is a similar spatial pattern for "Ginger", "Brookbank", and the unnamed tributary south of "Wasabi", differences exist for the main-stem channel erosion prediction. Background aerial imagery is public domain (Coconino County 2015) for Figure 7a, background imagery for Figure 7b is public domain USGS NLCD data (Homer et al. 2012).**

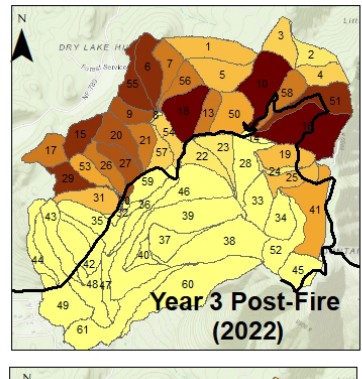

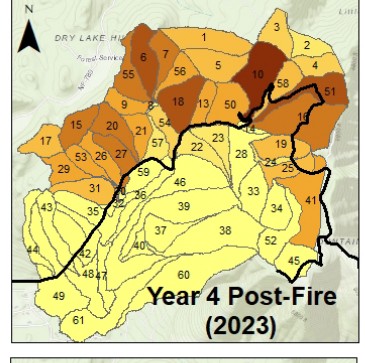

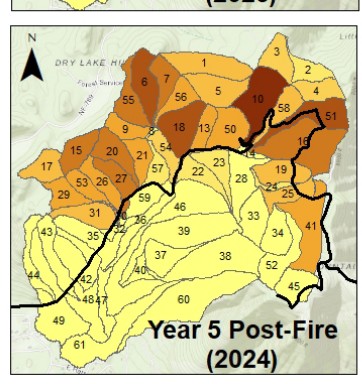

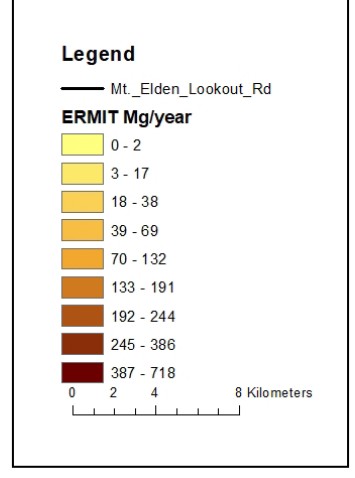

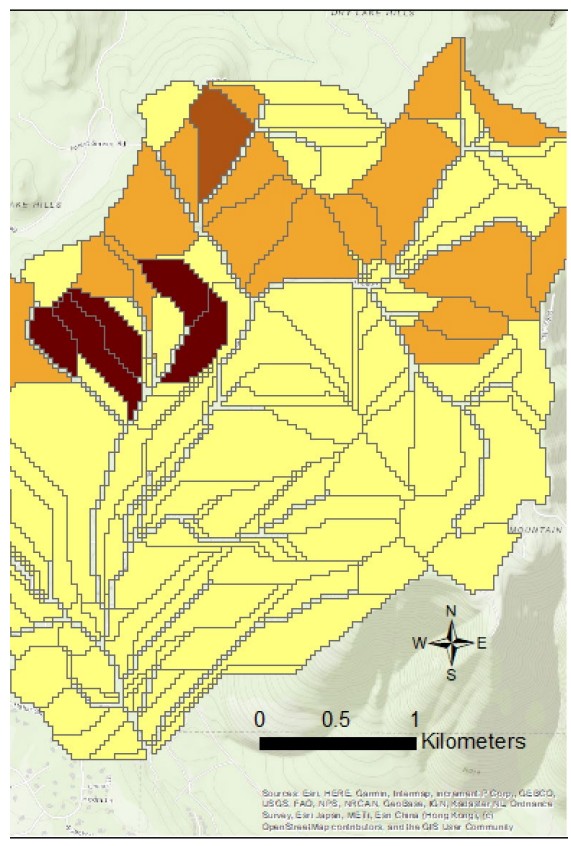

**A)**

**B)**

**Figure 8. Modeled hillslope erosion rates for 2022, 2023, and 2024 (A: ERMiT model) and 100 year forecasted annual hillslope annual yield (B: WEPP model). The WEPP model shading is to scale with the ERMiT model. Background imagery is USGS NLCD data in the public domain (Homer et al. 2012).**


### 4.2.2 Hillslope (MUSLE) Sediment Yield

The MUSLE model also estimates high rates of hillslope erosion for the four modeled precipitation events. The sub-tributaries utilized for the analysis are the same as those utilized for the average annual
sediment transport estimates from FLOWSED/POWERSED analysis. The results vary widely depending on the precipitation event utilized and the erodibility factor (K) of the soils. Based on field observations, the medium K value (0.545) likely represents the best estimate of aggregate soil conditions in the various watersheds within the burn area (Table 2).

A simplification of the observed 2021 rain events would provide a MUSLE sediment yield estimate of
4860 Mg, three 1-inch (2.54 cm) rain events in July and one 2 inch (5.08 cm) rain event in August; all medium K values. No other large rain events occurred in 2021 over the Museum Fire burn scar.

**Table 2. MUSLE model results for soil losses for three different soil erodibility factors (K) for three different rain events for 1 inch, 2 inch, and 3 inch (2.5, 5, 7.5 cm) in one hour. The medium K value is the most likely approximator for the 2019 Museum Fire. K values included 0.29, 0.545, and 0.80 for "low", "medium", and "high" based on soil conditions. See Figure 4 for location of Sub-watersheds.**

| Sub-Watershed Name | Soil Loss with low K value | | | Soil Loss with medium K value | | | Soil Loss with high K value | | |
|---|---|---|---|---|---|---|---|---|---|
| | 1" | 2" | 3" | 1" | 2" | 3" | 1" | 2" | 3" |
| | Mg | Mg | Mg | Mg | Mg | Mg | Mg | Mg | Mg |
| Ginger | 185 | 739 | 1455 | 348 | 1389 | 2735 | 510 | 2038 | 4014 |
| 40 Acre | 199 | 1055 | 2807 | 373 | 1983 | 5275 | 548 | 2910 | 7743 |
| Upper North Trib - Wasabi | 335 | 1868 | 4203 | 629 | 3511 | 7898 | 923 | 5153 | 11594 |
| Upper North Trib - Brookbank | 603 | 2941 | 6633 | 1133 | 5527 | 12465 | 1664 | 8113 | 18297 |
| Middle North Trib | 375 | 1956 | 4529 | 705 | 3677 | 8511 | 1035 | 5949 | 12493 |
| Lower North Trib | 196 | 837 | 1398 | 368 | 1573 | 2628 | 540 | 2309 | 3858 |
| Lockett Ranches | 367 | 2277 | 5793 | 689 | 4280 | 10887 | 1012 | 6283 | 15981 |
| Paradise | 197 | 1364 | 3596 | 370 | 2563 | 6759 | 544 | 3763 | 9921 |
| Park Basins | 192 | 1563 | 4272 | 361 | 2937 | 8028 | 530 | 4311 | 11784 |

4.3 Sediment Transport and Retention

Empirical results from in-city sediment removal, as measured at the Cinder Hills Landfill, are provided in Figure 9. The majority of sediment removed from the urban environment occurred after the first storms post-wildfire, during the July months (6260 Mg) with an additional 3760 Mg removed after the larger August flood event. All in-city sediment removed is downstream of the modeling regimes and

act as an end-point for the study system.

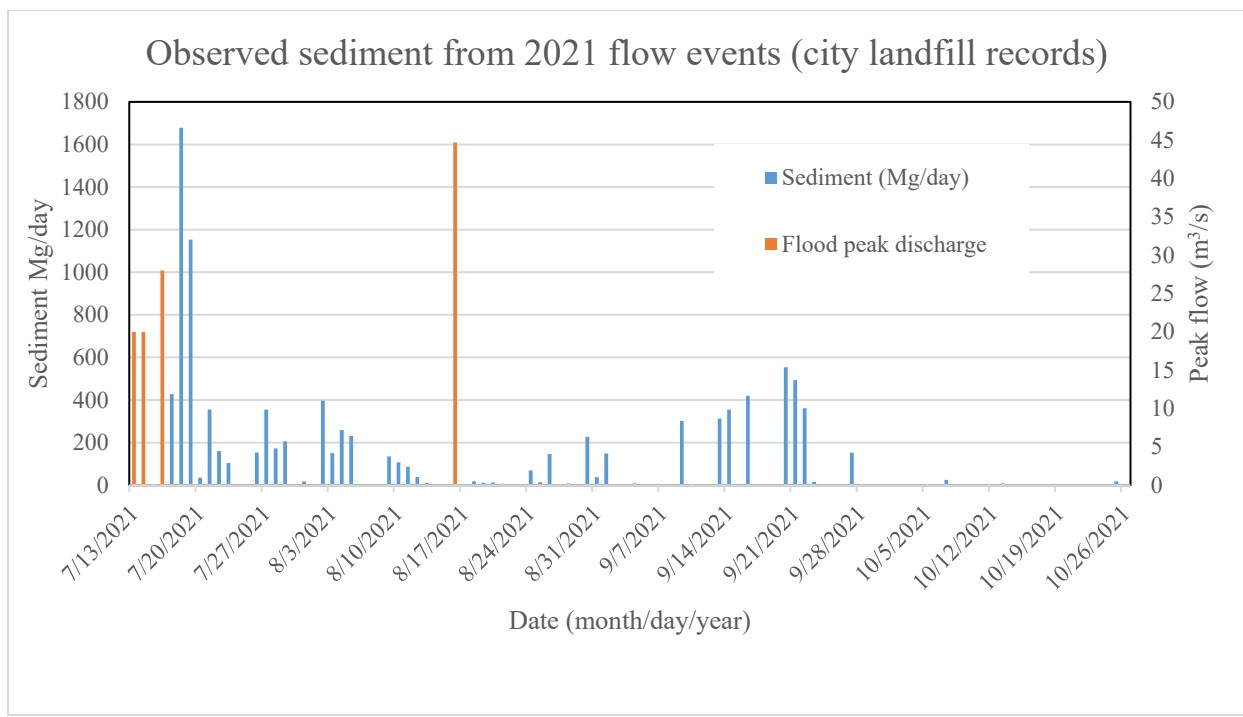

**Figure 9. Sediment and debris removed from channels and streets. Flood events occurred on July 13, 14, 16, and August 17th. Flood flows were predicted at the upstream entry to the CoF as 20, 20, 28, and 44.7 m3/s respectively (Schenk et al. 2023).**


The upstream FLOWSED/POWERSED sediment transport and on-forest retention modeling determined that five of the seven work area channel cross sections currently transport more sediment than is supplied to them, potentially leading to up gradient headcutting and continued erosion (highlighted in bold type in Table 3). These five proposed work area cross-sections transport sediment

more efficiently than the upstream sediment source cross section due to channel geometry, generally due to a headcut working into a "D" channel and converting it into a "G" channel. Once this process has begun, it exacerbates headcutting and fan degradation, channel migration, bank erosion, and provides little-to-no sediment aggradation (retention or deposition) on the now disconnected alluvial fan. Without direct intervention, these fans and channels will continue to efficiently transport sediment

downstream towards the residential areas.

Sediment transport modeling results (FLOWSED/POWERSED) indicate that design cross-sections retain an average of 70% of incoming sediment in proposed work areas than the degraded alluvial fans and channels in their current (fall 2021) condition (Table 3). It should be noted that large, single events are not modeled by this analysis and could potentially deliver more sediment. Flow events in 2022

were muted in Spruce Wash due to small rain events, the alluvial fan sites that were constructed prior to monsoon season did appear to function well in terms of sediment aggradation and attenuation (Figure 10a,b) however there were no flow events that over-topped the channel within the city to provide empirical comparisons. Field observations of similarly designed sediment retention structures on the nearby 2022 Pipeline Fire burn scar showed consistent sedimentation in the 70 to 80% range,

based on repeat surveys and sediment haul off from events in 2022 and 2023 (Tiffany Construction LLC and Coconino County Flood Control District personal communications; Beers et al. 2023).


**Table 3. FLOWSED/POWERSED model results indicating potential sediment retention for proposed sediment basins. Values in bold type are net erosional alluvial fans in the current (2021) condition, values in italic type indicates net aggradation (sediment storage). The annual sediment transport rate to the city neighborhoods is shown at the bottom in bold italic type (4628 Mg/year).**

| Potential Work Area Name | Incoming Transport Capacity | Current Channel Transport Capacity | Design Channel Transport Capacity | Difference between Incoming and Current Transport Capacity | Difference between Incoming and Design Transport Capacity | Sediment Retention at Proposed Design Channel | Percent Sediment Retention |
|---|---|---|---|---|---|---|---|
| - | (Mg/ year) | (Mg/ year) | (Mg/ year) | (Mg/year) | (Mg/year) | (Mg/year) | % |
| - | A | B | C | D | E | F | G |
| - | - | - | - | = A - B | = A - C | = E - D | = (1-C/A)*100 |
| Ginger | 377 | 324 | 54 | *54* | *324* | 270 | 78 |
| 40 Acre | 992 | 1326 | 269 | **-335** | *722* | 1057 | 66 |
| Upper North Tributary | 532 | 3055 | 215 | **-2534** | *317* | 2840 | 54 |
| Middle North Tributary | 1128 | 1363 | 399 | **-235** | *728* | 963 | 59 |
| Lower North Tributary | 1290 | 2341 | 441 | **-1051** | *849* | 1901 | 60 |
| Lockett Ranches | 1814 | 1426 | 762 | *388* | *1052* | 664 | 53 |
| Paradise | 2028 | *4628* | 450 | **-2599** | *1579* | 4178 | 71 |


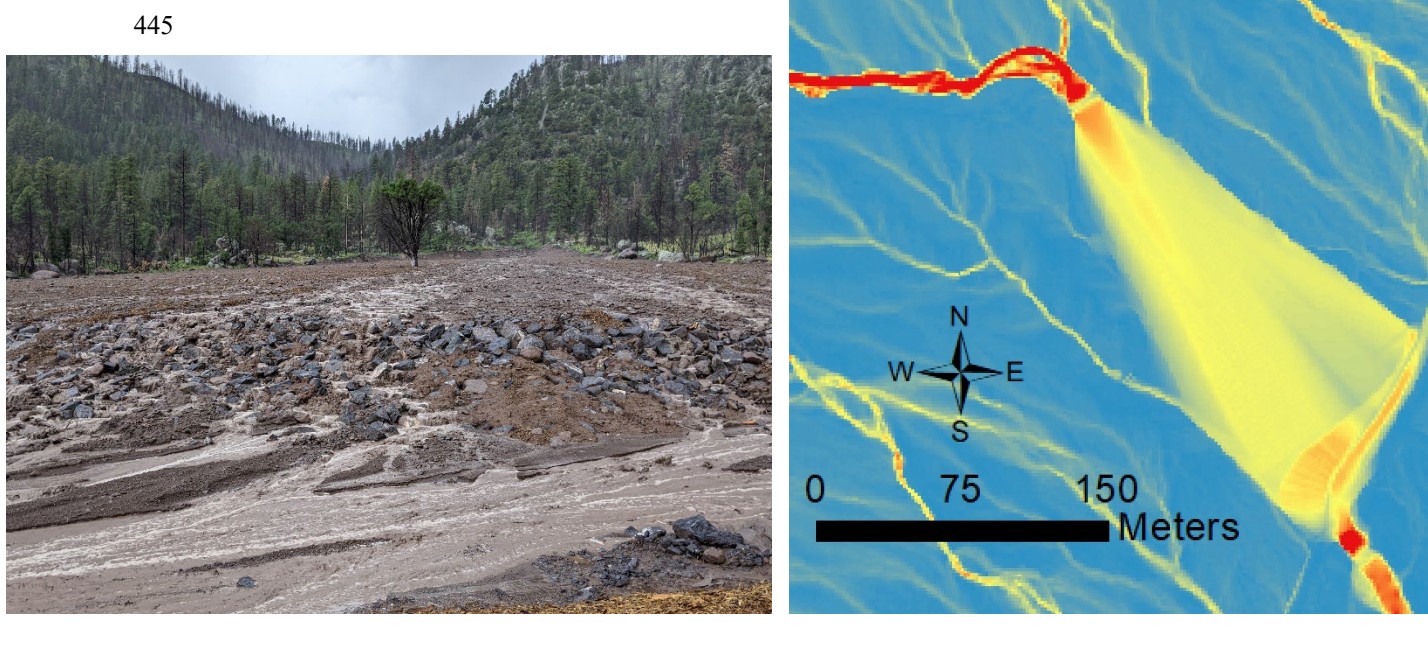

A)                                                                B)

**Figure 10 A) Ginger alluvial fan work site (35⁰ 15' 10", -111⁰ 38'0", looking upstream) during a July 2022 flow event, note the spread of flow and subsequent drop in water velocity allowing sediment aggradation.**
**B) Ginger alluvial fan work site including predicted velocities during a 25 year annual exceedance probability rain event. The dark red areas are approximately 3 m/s, the blue areas denote no flow (0 m/s). The photograph in Figure 10a was taken at the southeast corner of the fan looking northwest (upstream).**

The commonly used WEPP model demonstrated much lower sediment yields (3870 Mg/year) than the
WARSSS model (4630 Mg/year) and empirical results (9900 Mg/year in 2021) for the Museum Fire burn scar and Spruce Wash watershed and slightly less than the event based MUSLE model (4860 Mg/year).

# 5. Discussion


The 2019 Museum Fire, and subsequent nearby 2022 Pipeline Fire, demonstrated that previous hydrologic forecasts for watershed disturbance in the northern Arizona region were largely correct. Two studies of local ponderosa pine (*Pinus ponderosa*) forests indicated that current conditions exhibited complacent watersheds, but that the threat of wildfire would enhance runoff by one to two orders of
magnitude (Leao and Tecle, 2005; Quisenberry 2009), similar to observations in this burned watershed (Schenk et al. 2023; Sankey et al. 2024). Recent sediment risk predictions were also prescient, indicating orders of magnitude higher sediment transport post-disturbance (e.g. Neary et al. 2012; Natural Channel Design, 2012).

The four 2021 flooding events demonstrated a high sediment supply from the burn area with an observed
rate of greater than 9,900 Mg into the neighborhoods from the four flood events alone. The WEPP and WARSSS models appear to underestimate sediment delivery by roughly 50% based on empirical observations of the above average monsoon season of 2021. A portion of the underestimation of both modeling regimes is the lack of ability to anticipate hillslope gully incision. The reasons for the relatively large contribution from hillslope gully and rill erosion are not completely known at this time but are
likely partly due to the long period of watershed complacency in the San Francisco Volcanic Field (estimated at several thousand years; Stempniewicz 2014; Fulé et al. 2023) leading to abnormally large

amounts of stored hillslope and channel sediment at risk of transport after drought fueled catastrophic wildfires (Neary et al. 2012; Vanmaercke et al. 2021). The large antecedent sediment storage volume is not accounted for in WEPP or MUSLE and only partly accounted for in WARSSS through the empirical measurements used to inform BANCS. Other factors likely include uncertainty in the empirical estimates (both over-estimating due to water volume in the sediment/debris loads as well as under-estimation due to floodplain areas not addressed by flood cleanup efforts), as well as WARSSS and WEPP model limitations for rill and gully erosion processes (hillslope incision). Hillslope gullying is one of the most prevalent forms of erosion in Arizona post-wildfire environments making the estimation of their sediment yield vitally important (Neary et al. 2012). Other case studies have also shown that WEPP underestimates post-wildfire erosion, as does MUSLE (e.g. Fernández and Vega, 2018; East et al. 2021), there are still very few case studies of WARSSS for post-wildfire sediment modeling. The advantage of WEPP, over WARSSS, is its ease of use, free availability, and rapid learning curve, allowing for rapid spatial determination of high risk locations (Lew et al. 2022; East et al. 2024), however WEPPcloud does not incorporate ground truthed data, as is evident when comparing spatial "hot spots" of sediment yield in this case study. The WARSSS bank and hillslope predictions were informed by field measurements that largely corresponded with qualitative observations of spatial sediment yield.

All three modeling domains, MUSLE, WEPP, and WARSSS showed drastic increases in channel and hillslope sediment yields post-wildfire in this case study. Both WEPP and WARSSS predict slightly more sediment yield from existing channels than from hillslope processes. The similarity between model results, and less than an order of magnitude comparison with empirical results, indicate that both WEPPcloud and WARSSS are useful for sediment predictions. There has been some controversy about the use of "natural channel design" versus "analytical channel design" for applied geomorphology projects (e.g. Lave 2009; Kasprak et al. 2016), this case study suggests that both trains of thought have validity in the post-wildfire environment. Continued updates to post-wildfire sediment modeling has been called for by disparate studies at a global level (e.g. Lopes et al. 2021; Partington et al. 2022; Ebel et al. 2023) and the hope is this case study provides support for future improvements in the post-wildfire sediment monitoring, modeling, and applied mitigations arena.

Inthis study, most high erosion areas are identified high in the watershed. Steep slopes and lack of accessibility likely preclude active restoration of these channels or any hill slope activities other than revegetation by hand labor. Frequent debris flows, a separate sediment transport mechanism, also complicates restoration in the headwater steep slopes (Porter et al. 2023; McGuire et al. 2023). The nature of the channels (mostly G and F "Rosgen" type channels) indicate that the channel form is in the early stages of evolving to a stable form (Rosgen 2009). Formation of a small floodplain and reasonably stable channel side slopes (2H:1V minimum) will require the erosion of significant amounts of sediment. The process will likely take years to decades before relative stability has been reached (e.g. Hupp and Simon 1991; Montgomery and Buffington 1993; Jumps et al. 2022). As such, there is a potential for elevated sediment loading for the foreseeable future and subsequent elevated life and safety risk to the community (JE Fuller 2024; McGuire et al. 2024).

Several sub-watersheds were identified that exhibited higher hillslope erosion rates than adjacent channels. Initial post-wildfire sediment studies found that channel processes are generally larger sources of erosion, though that narrative is rapidly changing with more case studies and better landscape scale surveying and monitoring (Neary et al. 2012; Rengers et al. 2016; McGuire et al. 2024). The non-equilibrium hillslope conditions are cause for concern if they do not begin to improve soon as high sediment loads from hillslopes will generally contribute to further degradation of the receiving channel. Two consecutive years of drought likely contribute to this condition, however continued erosion and rilling hinder seed establishment further retarding recovery. The sediment transport models indicate a high potential for successful reduction in sediment as flows cross restored alluvial fan areas, this was observed in 2022 where observations at the nearby Pipeline Fire indicate a sediment retention greater than 70% on the completed alluvial fan projects within some of the impacted watersheds (personal comms. Lucinda Andreani, Coconino County Flood Control District Administrator) and similarly observed after the 2012 Waldo Canyon Fire in Colorado, USA which contained similar alluvial fan restorations (Rosgen and Rosgen 2015). However, there were some steep slope (>2%) alluvial fan work areas in Spruce Wash that performed poorly due to floods greater than the design storm (Beers et al. 2023; Rebecca Beers personal comms.). Some fan areas (especially the West Tributary or Ginger) have

the potential to not only reduce sediment transport but also sediment contribution from bank erosion. Current high bank erosion rates can be eliminated by eliminating the current gullied channel and restoring the fan function. Fan areas on the main channel of Spruce Wash which already store some sediment can be greatly improved by grading to restore the consistent fan feature.

The sediment mitigation structures, or "work areas", consist of an upstream, single thread "feeder" channel and a multithread anastomosing "fan" channel supported by lateral harden grade control (e.g. rock sills) before constricting back to a single thread channel to feed into the existing drainage downstream (see Rosgen and Rosgen 2015 for more detail). Sediment output from the restored fans appears to be moderate (approximately half of upstream sediment input) over a multi-year average

(authors' field observations). However, the relatively steep fans will produce higher shear stresses at high, infrequent flows.  For example, peak discharges modeled for a 2" (5 cm) per hour precipitation event over the whole watershed (~ 37 CMS, 4% annual exceedance probability) produce enough shear stress on the Paradise fan to move 30 cm diameter sediment.  Consequently, these infrequent precipitation scenarios will have the potential to move large quantities of material through the fan system,

even though most is retained on the fan.

# 6. Conclusions

The need for accurate, and rapid, post-wildfire sediment yield and transport modeling is evident by the increased role of wildfires in the wildland-urban interface and subsequent flooding. This case study

shows the utility of both WEPPcloud and WARSSS for predicting sediment transport to the city of Flagstaff, Arizona. The agreement between both models for sediment sourcing and transport, and within an order of magnitude comparison to empirical observations from flood events in 2021, is encouraging. The difference between models was largely in the spatial pattern of sediment yield. Both models indicated a slightly higher contribution from channels than hillslopes but WARSSS, because it is partly

empirically based, was better at identifying "hot spots" of both channel and hillslope sediment yield. Hillslope sediment yield nearly matched channels, indicating a high degree of hillslope gully and rill erosion, a process that needs further study. Continued advancements in post-wildfire sediment modeling will help inform managers and policy makers on sediment and flood mitigation strategies, planning, and design.

This study also introduced a post-wildfire sediment mitigation strategy through the restoration of alluvial fans. The fan "work areas" were identified using the POWERSED/FLOWSED sediment transport model, which was likewise informed by the sediment yield predictions of BANCs and ERMiT in the WARSSS model. Each restored fan work area included the removal of a single thread channel to a graded slope with lateral rock sills for grade control. These mitigations allow for the natural creation of an

anastomosing channel that drops out sediment due to the change in shear stress and stream power. Initial sediment transport model results indicate a reduction in downstream sediment transport of 70%. Ongoing monitoring of these mitigation structures is occurring both in Spruce Wash as well as in adjacent burn scar areas in Coconino County, Arizona, initial results indicate success during small to moderate flow events.

## Author contributions

Edward Schenk provided project administration, resources, visualization, data curation, funding acquisition, and writing. Alex Wood provided sediment investigation, formal analysis, writing, and data curation. Allen Haden provided methodology, project administration, funding acquisition, supervision, review, and resources. Gabriel Baca provided review and substantial editing, data curation, and formal

analysis. Jake Fleishman provided data curation and formal analysis. Joe Loverich provided data curation and formal analysis including hydrology and FLO-2D modeling.

## Acknowledgements

This project is a collaboration of many partners from the Coconino County, City of Flagstaff, U.S. Forest Service, U.S. National Weather Service, Northern Arizona University, and a great deal of assistance

from local hydrology and civil engineering consultants. Special acknowledgement goes to Lucinda Andreani, Coconino County Flood Control District Administrator, for her leadership during multiple fires and subsequent flood events in the Flagstaff area. Funding was provided by Coconino County general fund, Coconino County Flood Control District, and the City of Flagstaff Stormwater Fund. We would like to thank our journal reviewers as well as colleague informal reviews, comments and suggestions have made this manuscript much improved over our original work.

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
