# Peer review of "Edward R. Schenk1, Alex Wood2, Allen Haden2, Gabriel Baca2, Jake Fleishman2, and Joe Loverich3"

_EGUsphere, 2023_

## Author Comment (AC2)

**Response to Reviewer #1, January 18th, 2024**

**We would like to thank this anonymous reviewer for their feedback, the comments have led to a greatly improved manuscript. Below are the reviewer's comments in regular type and our responses in bold type.**

**General Comments**

This is a manuscript that describes several modeling approaches to estimate postfire sediment yields. They compare these yields with anecdotal data (photographs) and truck records that are subsequently used to estimate a sedimentation rate. The figures are generally clear, although somewhat repetitive, and thus there is an opportunity to reduce the current figures. The writing is generally clear, but some methodological details are missing, and some information is not well introduced in the paper (e.g., mention of the Pipeline fire) prior to the discussion.

I have several major suggestions to help improve the structure and clarity of this manuscript. The first suggestion is to better describe the type of model that you are using. Model is a vague term and can mean anything from a process-based numerical model to an empirical equation. Since many of these are previously published you don't need to get into all of the details, but I think you should describe the model type, the inputs, the parameters that you adjust, and the functional relationships. As far as functional relationships, that would mean that if, for example, you model deposition, then you say the key equation/method in which deposition is controlled.

**The model types are explained in more detail in this revision. However, the reviewer does correctly point out that there is abundant existing literature on the individual models. The purpose of this manuscript is not to introduce new modeling techniques but to compare existing models between themselves and empirical results. Adding too much detail about each modeling technique would detract from the purpose of this manuscript.**

Second, I think that the comparison between model output and observations could be clarified. Try adding a figure that shows a timeline of events that occurred. And then refer to the timeline when you say what you are modeling and how you are comparing models to observations. Right now this is confusing. In section 4.2.1 you are modeling four flood events, but were there more storms than that? Were those just the four largest storms? In addition to the timeline, I think you need to have some sort of figure that shows rain events as a function of time in the study. It is not always clear how what subset of postfire time you are comparing to the model.

**More description of the flood events (there were four) is provided in this revision. We could provide a timeline but that would be redundant with the explanation of events and the modeling domains. A new subsection, 2.1, was added to provide more clarity.**

Finally, organizationally, there are a few items that don't make sense. You start talking about the Pipeline fire in the discussion, but it wasn't mentioned prior to that in the study site section. Why do you model 4 events in 4.2.1 and three events in 4.2.2. How do the subwatersheds highlighted in figure 5 relate to "work areas" in Table 3? Should work areas be a single point rather than a watershed? And if not, then I think you need to state that a work area is a full subwatershed. And there could be an overall improvement of descriptions. For example, in the caption for Figure 10 it says "note the spread of flow and subsequent drop in water velocity", however you are not actually showing anything that indicates velocity. Other examples can be seen in the specific line comments below.

**These are good comments and are addressed in the line comments (which are the same as the paragraph above). Figure 10 has been updated to include a velocity map.**

**Line Comments**

1. Here you use "post-wildfire" but on line 50 you write "post-fire". Try to use a consistent term throughout the manuscript.

**We have replaced "post-fire" with "post-wildfire" throughout the manuscript, thank you for catching that discrepancy**.

1. Paradise/sunnyside is not shown on the map in figure 1. Only Paradise is shown. Can you fix this?

**This is fixed in this iteration of the manuscript, thank you for catching that omission.**

1. Add a reference at the end of this sentence to support the statement

**Unsure which sentence, missing a reference to the line number in this comment form.**

1. after the word "debris" can you reference a news article to support this?

**A link to the NOAA webpage for the flood events (https://www.weather.gov/fgz/FlagstaffJuly2021) is now included.**

1. when you say "increase in runoff" what is the baseline for the increase?

**"Surface water" has been added for clarity. The basis for the modeling and empirical observations of flooding are provided in the reference in the sentence (Schenk et al. 2023).**

113-114. When you say that the watershed was divided into sub-watersheds, say the criteria you used for the watershed delineation.

**"Based on USGS National Hydrography Dataset" has been added. With a citation of:** USGS, (U.S. Geological Survey), 2019, USGS 3D Elevation Program Digital Elevation Model, accessed October 31st, 2021

1. Sentence ending on 127, support claim with a reference.

**More description of the highly heterogeneous rainfall distribution in the American Southwest was added to this sentence.**

128-139: Why is this paragraph not in section 3.4?

**This can be explained either in the Sediment Modeling section or the Sediment Yield section, providing the method twice is redundant.**

1. I think you need to add some details to explain how "an analysis of sediment transport across a conceptualized design channel" was done. This is really vague.

**We took cross sections in their condition at our proposed fan work areas and modeled them as 2% consistent grade as a proposed condition. We compared the modeled cross section with the current 2021 condition.**

1. Say how these data are used to estimate bank erosion. What is the method?

**Bank erosion was quantified in the field in 2021 and estimated in out-years using channel slope, expected annual flow rates (from FLO-2D modeling completed outside of this study; Schenk et al. 2023), soil condition, and channel cross-sectional area.**

185: I think you mean model instead of modeled

**Thank you, corrected.**

189: State explicitly why you considered 2021 as the second year post fire.

**The fire began during the monsoon season of 2019 which would make 2020 the first full year post-fire.**

Say why you chose one inch, two inch, and three inch precipitation events. Also say the duration? Is this over an hour, a day, etc?

**"in one hour" was added to provide the appropriate detail.**

1. Say why you are using those K values. Also, did these vary by year?

**We chose those K values based on field observations. The MUSLE modeled watersheds were larger than the ERMiT watersheds (because we were looking at different metrics and different timescale...annual average sediment yield vs event-based sediment yield). The condition of the soil pre fire was sandy loam with a modeled K value of approximately 0.29. Post fire, we anticipated that the K value was ~0.80. Averaging these two balanced the initial condition vs the post-fire condition.**

1. Did your c value vary by year?

**No, it did not change by year because a) it is a dimensionless number used to denote a landscape scale practice (in this case it was forest area, doesn't change per MUSLE year to year)) and b) because we only completed MUSLE for current conditions as an event-based modeling method (not annual for multiple years.**

1. State error associated with photo measurements.

**The manuscript explains that these were qualitative assessments, there is no known error to the qualification nor are there repeatable confidence limits for this method. The landfill scale measurements were over several weeks and likewise do not have confidence intervals or a standard for error measurement.**

1. I think you actually need to describe the model details here. Is this just an equation, a numerical simulation, a spreadsheet with a GUI wrapper, something else? Show what calculations are being performed.

**This section has been expanded to describe the modeling technique and inputs.**

**The following was added to the manuscript:**

**"FLOWSED models the total annual sediment yield, both suspended and bedload, using flow-duration curves and their corresponding sediment yields. The dimensionless flow-duration curve is developed from representative watersheds**

**in the region using USGS stream gage data. The POWERSED model compares sediment transport in various configurations of channel geometry."**

1. Say how FLOWSED/POWERSED estimates rebuilding of alluvial fans. What equations/methods are used to estimate deposition.

**FLOWSED/POWERSED are sediment supply and sediment transport models. They are used to determine the amount of sediment moved in a year by the channel. In this case a narrow, gully channel with no floodplain (existing condition) is compared to a valley wide channel (proposed alluvial fan). The difference in sediment transport is compared across these cross sections to show how the sediment transport competency changes with channel geometry.**

1. Where channel types updated? If they were not, why not?

**The channel types presented in Section 4.1 were assessed after the 2021 flood season and already account for changes in post-wildfire geomorphology.**

1. Sqy what you used to estimate the transport capacity and then how that was taken into account.

**This is addressed in previous comments on the FLOWSED/POWERSED modeling.**

1. You looked at 2021 but used 2022 modeled data. Explain why here.

**We took the current condition during the fieldwork period (Fall 2021) and used this period of time to model for 2022. The model uses the stream banks and hillslope in the current condition, considers average annual precipitation for the calendar year, and then models sediment output for a year. Since we completed the analysis and modeling in late fall 2021 (after the 2021 monsoon season), this analysis and modeled results are predictions for sedimentation for the entire 2022 winter runoff season and 2022 monsoon season. We didn't use other years (2019 or 2020) because we didn't have relevant field or geospatial data for these time frames.**

1. Change "calculates" to "estimates"

**Done**

1. Cite "a table or figure after "erosion rates."

**Figure 7 and Figure 8 are provided, and cited, for displaying the data from this section.**

1. Say the reasoning behind the different precipitation events you modeled in the methods. Also, state why you didn't just use the observed precipitation.

**The reasoning has been explained in the Methods for each of the models as well as using the four observed flood events.**

330-331. Is this the full inventory of storms? I think you need to actually show the storms that occurred, and so I'd suggest creating a new figure showing that.

**A sentence was added to clarify that these storms were the only ones to generate flood runoff in 2021 within the study area.**

1. This sentence doesn't make sense here. Topic sentence of paragraph is about precipitation, but here you switch to a totally different topic of sediment removal.

**This sentence was moved to the next section on sediment transport and retention.**

1. here you are talking about a result, but you don't mention the mechanism that triggers more sediment retention?

**More description is added to this section.**

356-357. Here you talk about 2022 flow events. This makes me think that it might be helpful to have a timeline so we can better keep track of events that you are investigating.

**The sentence explains that the 2022 flow events were minor with no flooding. More context is added to this section. During the study there were only four large flood events, which are described in the manuscript.**

1. Sentence that starts with: "Observations on …" doesn't seem relevant. What about this statement makes it go with the rest of the paragraph? It seems like a random statement about a different fire.

**Added context, the Pipeline Fire watersheds had similar sediment retention structure designs ("alluvial fan restorations" or "work areas").**

1. Here you start talking about the Pipeline fire again. But this is not mentioned in the study Site section or the methods. If you want to talk about it in the discussion, it needs to be introduced in those sections beforehand.

**The Pipeline Fire is not part of this study, discussion of a nearby fire, outside of the study, is appropriate in the Discussion section. Adding external discussion**

**pieces to the Study Site Section or Methods would not make sense. A description of the Pipeline Fire has been added to the Introduction.**

1. Did you see 1-2 orders of magnitude change in runoff? Say how your observations were similar or different than this prediction.

**Added a reference to hydrologic observations from the Museum Fire watershed.**

1. Reference a table or figure after 2021 on this line.

**The flooding events were described previously in the manuscript.**

1. Define "complacency"

**Complacency has been indirectly defined already in the Discussion, more information is available in the references provided (Stempniewicz 2014; Fulé et al. 2023).**

Be more specific about what empirical estimates and "other factors" you are talking about here.

**Unsure what specificity is requested. The sentence already provides the assumptions and unknowns that might lead to error: "Other factors likely include uncertainty in the empirical estimates (both over-estimating due to water volume in the sediment/debris loads as well as under-estimation due to floodplain areas not addressed by flood cleanup efforts), as well as WARSSS and WEPP model limitations for rill and gully erosion processes (hillslope incision)."**

420-421. You don't talk about different channel designs in your methods so it is out of context to bring this in now.

**The Discussion section is the perfect location to provide a discussion of the results, techniques, and methods. The "Rosgen" style of channel design, also known as Natural Channel Design, is commonly used in the USA but is not without its controversy and detractors. Lines 420 through 429 provide discussion on the use of this channel classification and design method in this study and how it compares favorably with a more generic WEPP model as well as empirical results.**

1. What evidence suggests that the channels are evolving to a stable form?

**The sentence in line 435 is prefaced with the channel type and based on "natural channel design" process interpretation. A reference has been added to the appropriate paper explaining this process (Rosgen 2009).**

1. Here you suggest eliminating the gully, but one popular definition of a gully is that it is not a feature that can be easily reworked by machinery. Once established it is likely to come back.

**A sentence has been added explaining the use of rock sills as grade control to maintain the alluvial fan plan view (lines 457 to 459).**

1. Say the recurrence interval.

**Line 454 now explains that this a 4% annual exceedance probability**

**Figures and Tables:**

Figure 1. Make the tick labels bigger. If you are running out of room on the latitude tick labels, you can rotate them. Label Spruce Wash. Use a (b) to label and refer to the inset. And in the inset say what the shapes represent, I assume they are county boundaries? Add a polygon showing the fire perimeter. Also the line type for the forest boundary and Mt. Elden Lookout road are difficult to distinguish.

**Figure 1 has been updated.**

Figure 3. I think figure 3 can be combined with figure 1 because most of the information is the same. Also, I see some cross sections near the label for "Lower North Trib" that aren't crossing the stream lines. Why is that?

**We left Figure 3 in "as-is" since it would make Figure 1 very busy and difficult to read to add the cross-sections on top of the other neighborhood labels and delineations of the sub-watersheds. The cross sections may not line up with the stream lines if the National Hydrography Dataset (NHD) has mapping errors. The NHD is the base layer for the stream lines shapefile.**

Figure 4. Can you put the alphabetical labels close to some of the stream segments on the map to better see how the segments are related to the legend?

**We have updated the figure.**

Figure 5. Say what sets the categories for unit erosion

**Unsure what the reviewer is asking for, the units are presented in the figure and the figure legend.**

Figure 7. The labels in Figure 7a are too small.  In caption, consider changing "major channels" to "tributaries"

**This figure has been improved accordingly.**

Figure 8. In the caption say why you chose the 100 year forecasted hillslope annual yield

**This is a common WEPP model output and is explained in the WEPP model literature cited in this manuscript.**

Figure 9. Your arrow is attempting to indicate two things. So I suggest you actually swap the vertical error for a vertical bar. Then have a horizonal arrow going in one direction to indicate 6260 Mg before August 17, and a second horizontal arrow in the opposite direction to indicate 3760 Mg after August 17.

**Unsure of what the request is for this figure. The current figure shows the amount of sediment removed from the neighborhoods by day and also delineates the difference between the first three storms and the larger August storm.**

Figure 10. make a figure 10 showing the observed precipitation.

**The four flood events are described in the manuscript, a more in-depth discussion is provided in the cited hydrology report (Schenk et al. 2023). The study area is in a semi-arid climate with infrequent rain and flow events. Flow events that did not trigger a flood are extremely minor with little sediment mobilization.**

Table1. Add name of model in column labels

**The model used for each column is already expressed in the table caption. Adding this detail, and another explanation, in the table would be duplicative and make the table less concise.**

Table 2. Put the K value in the column titles.

**Added:**

**K-Values:**

**Low K: 0.29 (sandy loam)**

**Medium K: 0.545 (sandy loam + high burn)**

**High K: 0.80 (High burn severity K Value)**

Table 3. Indicate a column title for the last unlabeled column

**The header has been fixed**

---

## Author Comment (AC3)

**Response to Reviewer #2, August 8[th], 2024**

**We would like to thank this anonymous reviewer for their feedback, the comments have led to a greatly improved manuscript. Below are the reviewer's comments in regular type and our responses in bold type. We expect to update the manuscript to the EGU website after this Response to Reviewer.**

Schenk et al investigate sediment source and their transport in a region of Arizona that has been impacted by wildfire. Their work is strongly framed around modelling approaches and comparisons. They also use empirical information to validate their model outputs and discuss mitigation efforts. Considering the context of such processes with respect to the occurrence of landslides processes such as debris flows; the present study is relevant to the audience of NHESS. However, I think that there is room for improvement, notably via an effort at presenting this work for a broader audience. Too often the manuscript reads as being very case-study focussed and the connection with other studies (either modelling or result based) is missed.

Overall I also agree with the comments of the first reviewer who points out to several key aspects. I have additional specific comments to hopefully improve the manuscript.

**Thank you, please refer to our responses to Reviewer #1 from January 2024. We improved and clarified the manuscript based on their comments and suggestions.**

**Abstract:**

The abstract sounds very technical, especially for readers that are not directly familiar with the models. Abbreviation are usually to be avoided here. Quantitative values, if mentioned, should ideally be compared/discussed with the broader literature.

**We understand the need to avoid abbreviations in the abstract but unfortunately these model name acronyms are analogous to the model name (nobody references these models by their full names and each of these are discussed widely by their acronym as a surrogate for the model name). Attempting to write out the full model names in the abstract would be cumbersome and reduce the readability. The quantitative values provided in the abstract are directly comparable between observed rates of sedimentation and the different models. Providing a comparison with the broader literature, in the abstract, would not be practical since there are multiple variables that go into an event-based sediment laden flash flood event. The purpose of the paper is to discuss the precision and**

**accuracy of different sediment models and not provide a comparison of sediment flux between disparate studies of post-wildfire runoff. The numbers provided in the abstract are directly comparable between different models and the observed sediment deposition.**

**Introduction:**

Overall, this section could be improved thinking about the broader audience. It needs to provide a better justification of the models used. The study area must also be better justified. . See specific comments below:

Lines33-34: the focus seem to only concern the American West. I would welcome an introduction that goes broader. In other words, can a researcher from Spain, Greece or Mexico for example be interested in this research as well?

**We changed the sentence to be more inclusive to humans living at the WUI for semi-arid forests anywhere globally. At the time of the initial writing the WEPPcloud model was only available for the USA, it has now been expanded for soil classification databases on multiple continents making it of a more global use.**

Line 60: three models acronyms (please explain the acronyms when mentioned the first time) are introduced without any justification. As it is the introduction, these models should be backed-up via the state of the art.

**A sentence is added in the Introduction that provides the long form name of each model, the discussion of the modeling techniques is already included in the Methods including the rationale for using each of the techniques. We feel this justification is more applicable to the Methods section than the Introduction.**

Line 21: the study area is mentioned without any broader context. In other words why is that study area of interest for the international audience. Why is that an ideal case study? The goal here is not to repeat what is proposed in section 2, but instead make sure that a reader from, for example, China, finds Flagstaff a place of interest.

**This comment could be said for any study. We revised Line 21 to further explain the context of the fire area and study (removed the name of the watershed and explained the geologic underpinning of the burn scar).**

**Study area:**

make sure that all the local names are relevant. Overuse of such names are not ideal for the understanding of the research.

**We agree, however the use of local names in the Study Area section is intentional, it provides needed context if a researcher was to attempt to replicate this study or confirm the work that we are presenting. If a reader finds the detail distracting they can read ahead to the next section.**

**Method:**

Overall, there is a lack of method justification with respect to the literature. See specific comments below.

Line 104: what does the acronym FLO-2D stand for?

**While written like it is an acronym it is the actual name of the model, a search of the internet and a literature search on Google Scholar did not find a different name other than "FLO-2D".**

Line 105-110:  What Lidar data are used? Provide source, resolution information, etc.

**Both lidar datasets (2015 and 2019) are available on the USGS National Map server. This information has been added to the manuscript including a link to the National Map server. The horizontal and vertical resolution vary between both datasets but are sub 10 cm accuracy or better.**

Why (based on what physical criteria?) these grid scales in the flood modelling ?

**The initial grid size was selected based on emergency conditions during the wildfire (the ability to provide quick results to the emergency management teams). The more refined grid element in 2019 was selected to provide a better resolution of rainfall-runoff. The relative trends for sub-watershed runoff did not change with the change in grid size. The overall watershed is sufficiently large that either grid size provided relatively precise results for runoff and flood events.**

Lines 151-156: can you clarify on the sediment transport analyses carried out? Can you the back this up with literature?

**This is a similar question to Reviewer #1, we have improved the manuscript after those initial comments to better frame how FLOWSED/POWERSED operates and how it has been used previously in the scientific literature. Thank you for sharing**

**a similar concern as Reviewer #1, we hope the revised text provides needed detail. Please refer to the Response to Reviewer #1 for more information.**

Lines 184-186: any reference for these modelling approaches?

**References to both modeling approaches are included in the manuscript.**

Lines 200-205: any of the values used in the model can be justified from the literature?

**The values used for the MUSLE model were predicted based on field conditions, this is now clarified in the manuscript. More information about determining the K value, C value, and P factor are available in the reference provided in the manuscript.**

Lines 207. CoF staff?

**CoF is defined on Line 82 (City of Flagstaff).**

Lines 207-212. Reference(s) to support these methodological choices?

**The rationale for the methodology of the empirical measurements is included in Line 212 (observations were noted for Federal and State disaster declaration reimbursements). Due to the emergency nature of the urban flooding there was no opportunity to collect more holistic measurements using precision surveys. Cleanup operations commenced during the falling limb of the flood, the most appropriate way to capture sediment deposition in the urban environment was the Federal and State disaster reimbursement paperwork submitted from the landfill and through flood observer photos.**

Results:

General comment: can the erosion/sediment values obtained in this study be compared to other cases? That could help to make the discussion even more interesting/of a broader interest. Providing quantitative values without putting them into perspective is not always relevant.

**The purpose of this paper, as stated in the Introduction, is to provide an example of three different sediment modeling techniques and compare the precision and accuracy with empirical results. Comparing gross sediment values to other studies would not be very interesting based on the number of confounding variables that make comparison difficult. This could include the number of flood**

**events, rainfall hyetographs, burn severity comparisons, slope, geologic provenance, watershed recovery/vegetation recovery, and other variables.**

Lines 395. The author refer to gully erosion. This comes as bit as a surprised that this process is not mentioned earlier in the study area section. Something that remains unclear is the origin of these gullies. Where those gullies be already in place before the wildfire? If so, would these gully be associated with earlier wildfires? In some cases, gullies are not to be the consequence of landslide processes. Are their observation of landslides in the regions. Landslides could develop after wildfire of course, but could also be there as a basic geomorphic agent that bring sediment to the river system. Overall, some clarification (extra relevant information) around these mass movement/erosion processes would be welcome is that helps to better understand the model outcomes.

**I believe there may be a definition miscommunication here. The hillslope gully erosion mentioned on this line is for the formation of hillslope gullies and rills through hillslope erosion. We are not talking about mass movement or mass wasting. No mention of landslides or debris flows is provided in the manuscript in terms of modeling or empirical observations. There are plenty of examples of gully erosion definitions in various government reports from agencies on multiple continents, this manuscript subscribes to the common definition of the term and is discussed at length in the Discussions section. Mentioning gully erosion as part of the hillslope erosion modeling is described in the Methods section.**

**The onset of gully erosion is interesting in this region due to the prevalence of gullying on hillslopes post-wildfire. The process is more fully explored in an earlier paper from the nearby Schultz Fire that is cited already in this manuscript. The reference for that paper is as follows:**

**Neary, D.G., Koestner, K.A., Youberg, A. and Koestner, P.E.: Post-fire rill and gully formation, Schultz Fire 2010, Arizona, USA. Geoderma, 191, pp.97-104, 2012.**

**Another reference to the Neary et al. study has been added at this line location.**

Note here a reference of gully erosion modelling. Although targeting different scales, that could be useful: Vanmaercke, Matthias, Panos Panagos, Tom Vanwalleghem, Antonio Hayas, Saskia Foerster, Pasquale Borrelli, Mauro Rossi et al. "Measuring, modelling and managing gully erosion at large scales: A state of the art." Earth-Science Reviews 218 (2021): 103637.

**Thank you for the reference, the paper is interesting and is now included in the Discussion section under the gully conversation.**

Figures:

Figure 1: add elevation quotes.

**Unsure of where the reviewer would like elevations called out, the topography is highly heterogenous. Example elevations for Mount Elden, Dry Lake Hills, Mount Elden Estates neighborhood, and Paradise/Sunnyside neighborhoods are now included in the Study Site section. Contour lines, and DEMs, for the area are freely available online at multiple sources (e.g. USGS National Map, Google Earth, City and County GIS portals, etc).**

Figure 2. Indicate when the photos was taken. Provide also the geographical coordinates of the photo.

**A year and season is now included in the figure caption as is an approximate location that can be compared to the site map.**

Figure 10. Indicate when the photos was taken. Provide also the geographical coordinates of the photo.

**A month was added to the existing year in the caption, the geographic coordinates for the "Ginger Fan" were added to the caption.**

---

## Referee Report (RR1)

**Comments on Revision**

The authors have addressed some of the comments from the last revision, but several major concerns remain with the existing manuscript. First, some of the literature citations are inappropriate. For example, they discuss sedimentation modeling and use a reference to a hydrology-only paper that does not address sedimentation directly (see line comments below). Second, the authors failed to address the comment about a timeline of events, and this remains important. This could be done relatively easily with a few additions to Figure 9. For example, in addition to showing the sediment, the authors could simply add a secondary axis with rainfall intensity that shows the timing of the major storms. Crucially, I think the authors need to also show the timing of the completion of their "alluvial fan restoration areas" on Figure 9 as well. The third and final concern, is that the authors do not take the opportunity to explicitly compare observations with modeling. For example, everything in Table 1 appears to be a modeling result, but they have an opportunity to compare the model with observations.  Same comment with Figure 9. That shows sediment observations, but it'd make sense to compare these with the modeling. I realize that the modeling and observations are obtained over different timescales, but you can sum the observations to the annual time scale and compare that to the modeling. Same comment with Table 3. Why not add a column that says "Sediment Retention Observed at Channel Design".

Right now my biggest concern is that this paper is a site-specific case study that does not have large application outside of the specific scenario. Moreover, I'm concerned that because the modeling is not compared to observations directly it's hard to evaluate how well these models actually estimated what was observed. Finally, I do not understand how the mitigation structures influenced/did not influence the sediment retention because I do not know when they were built with respect to the storms.

**Line comments**

29 I understand what you mean by sediment forecast here, but I don't understand what you mean by sediment forecast on line 28, where you say that "Sediment mitigation structures … are discussed as real-world applications of sediment forecasting…". I think this might just be a problem of language imprecision. I think you mean something closer to: "We discuss the real-world implications of using models to make sediment forecasts and using modeling results for the design of sediment mitigation structures." Either way, please refine this sentence because right now it sounds like you are saying that sediment mitigation structures are provide sediment forecasts, which doesn't make much sense.

34 Provide references to support this assertion, especially references that indicate that it is an "increasingly important issue", or change the language if that isn't something that is supported by research.

36. Sankey et al., 2017 is not a paper about flooding, and therefore you should adjust the reference. The word flood is not even used in that paper.

40. Ebel et al., 2023 (which doesn't have a year in your references) is about hydrology and does not specifically address these concerns: "damaging debris flows and sediment sourcing, transport, and aggradation". Here and elsewhere, please use references that support the assertions you are making.

46. Same problem as the last two. These references aren't really well suited for the points you are trying to support.

51. Either I'm confused about what you are referencing with Ebel et al., 2023, or you are confused. But I think you are trying to refer to this paper that is specifically focused on hydrological modeling and does not touch on sediment modeling at all. Is there a chance you are actually referring to a different paper?:

Ebel, B. A., Shephard, Z. M., Walvoord, M. A., Murphy, S. F., Partridge, T. F., & Perkins, K. S. (2023). Modeling Post-Wildfire Hydrologic Response: Review and Future Directions for Applications of Physically Based Distributed Simulation. *Earth's Future*, *11*(2), e2022EF003038.

62: Here you reference a conference abstract by Beers et al. 2023, and there is no mention of a loss in stream power or accretion upstream of neighborhoods in that conference abstract. Also, you inadvertently changed the title of that conference abstract, so I suggest you change it back to the original title. In any case, I don't think that the information in that abstract can be used to support the assertion here.

103 When you say that the flood events allowed for empirical comparisons to the modeled predictions be more specific. Do you mean you compared sediment discharge, volume, mass, flood velocity, etc? Specify to readers what exactly could be compared.

119: a 10-100 time increase in surface water runoff compared to what? Mean annual flow? Also, by runoff, do you mean discharge? Or depth? Be specific here.

126 There is something I don't understand about this: "Areas downstream from high sediment yield areas were identified as "work areas" …". I thought that the "work areas were specific mititation areas that were defined. This makes it sound like any area downstream from any other area with a high sediment yield is a "work area". I think you need to clarify this, and locate these areas on a map.

148: International readers of this European journal are likely not concerned about "who" estimated discharge, but I think they will be concerned about "how" it was estimated. So please include that information.

152: Are you saying that the floods incised the channels and then they will widen over time. Or are you saying that they were already incised prior to the fire, and will widen due to flooding. Please clarify.

154: here and elsewhere, when you are describing methods you have used, please use past-tense. For example, you say: "Sediment transport estimates are …", but that activity is over, so you should say "Sediment transport estimates were …". Same comment on 165 "sediment transport analyses are…" should be "sediment transport analyses were…"

207: Here you need to say the version number of the software, and provide a reference in your works cited.

214 say why you chose 1 inch, 2 inch and 3 inches per hour. Also, use "in" rather than a quote to abbreviate inches, as this will be more clear for an international audience.

223 I don't think you can use these words together " measured qualitatively" because if it's qualitative it's not actually a measurement. So change to something like "estimated qualitatively". Also, specify the difference between sediment and debris here. Is debris wood and trash? Say how it's different than sediment, or just remove the term debris.

234 after "…average annual sediment transport…" add units (e.g., Mg/yr) so readers know the units you are using.

243-244 I don't think you can justify this choice. In alluvial rivers the two-year return interval is often substituted for Bankfull, not the one year RI. Also what do you mean by "post-wildfire channel forming discharge" I don't think that's a concept that's been shown. What would be different about a postfire channel forming discharge versus a non-postfire channel forming discharge? Why is "channel forming" relevant here? If you think these choices have merit, then I think you need to explain them or point to literature that explains them.

257: You need to provide this information in a table somewhere and then reference it here: "bankfull cross- sectional area, Manning's n value, bankfull discharge, slope, suspended sediment (mg/L), measured bankfull bedload (lb/s), a flow duration curve, and a sediment rating curve comparison"

263 I'm a little lost on how you would use a "bankfull flow" if you are trying to estimate flow on an alluvial fan. Please explain.

285: Insteady of saying "G" channel, which is not a universally understood metric. I suggest you use a few short descriptions of the channel type. Something like "the channels are riffle-pool channels defined as "G" in XYZ scheme".

306 Avoid contractions like "don't" in scientific writing

367 Earlier you said there were four events. Please correct or explain the discrepancy. On line 24 it says "MUSLE predicted 4860 Mg/year (based on the four events)"

374 Again the reasoning for these choices of rain events were not justified. Please state why you chose those particular storm amounts/durations.

398 State if field observations confirm the modeling here. I think that readers will want to know if the modeling matches what was observed at those cross sections.

408 I think the critical question here is how do the modeling results compare to field observations. You have an opportunity here to compare model predictions with field observations, and that's the critical piece that seems to be missing.

416 This mention of the Pipeline needs more detail. How was the 70-80% measured? Was it repeat lidar, photogrammetry? Also, I think a critical piece is the timing after fire. Retaining 70-80% of sediment several years after the fire doesn't really say much about how the sediment retention structures would behave immediately after the fire. It is well known that wildfire sedimentation is typically highest in year 1 and goes down drastically in the following years as the watershed recovers.

450 What "hydrologic forecasts" are you referring to here?

501 I don't think that prior research supports this assertion that there will be "...substantial sediment loading for the foreseeable future..." it is well known that sedimentation rates after fire decline precipitously after the first year after wildfire. Within 3 years it is very unlikely that you'll continue to have fire-related sedimentation problems. Provide evidence that would support this point if you want to make this assertion.

507 Here and elsewhere, suggest replacing "poor conditions" with something else (e.g., transient or non-equilibrium conditions). I think you are trying to say that there is erosion or change, but calling it "poor" implies a judgement call on what is good/bad, which is really dependent on the observer.

514 Again provide evidence of how you measured 70% sediment retention on the pipeline fire. Also, I just looked up the Pipeline fire and I can see that it burned starting in June 2022. Did you have alluvial fans built on that fire to capture sediment in 2022? If so state the date of when those sediment retention structures were built with respect to the date of the fire and storms.

526 Again, the Beers et al., 2023 reference is just a conference abstract so there is no additional detail to be found on this.

527 Quantify what you mean by "moderate" and "long-term average"? Do you mean you retain 50% of the sediment or 98% and is this 50% of the long-term average measured using some dating technique or something else?

Figure 4. Somewhere, maybe in a supplement. Provide definitions for all of these channel types.

Figure 9. Please add rainfall intensity as a secondary axis on this so readers can compare the sediment mass per day with the rainfall intensity. You mention the four storms, but it'd be helpful to see where those storms exist in time compared to the sediment yield. Please also indicate the date that the alluvial fan restoration areas were completely constructed. You can do this with a single vertical line on the completion date.

---

## Author Response (AR2)

**Authors' Response to Comments**

November, 2024

Below are the authors' response to a second round of reviewer comments. The original comments were reviewed by the authors and a response was submitted to the journal on August 2024. There are some new comments that continue to improve the manuscript but there is also some confusion as to the goal of this study which leads to reviewer comments that are less beneficial. The purpose of this study is to assess the precision and accuracy of three commonly used sediment modeling techniques for post-wildfire mitigation. The study used empirical field observations to make this assessment possible. The results are useful for evaluating the utility of each of the models as well as for determining areas for improvement (e.g. hillslope gully process is not evaluated by WEPP, MUSLE was not originally intended for non-equilibrium post-wildfire conditions, and WARSSS relies on a proprietary set of models that provides similar results as WEPP at the watershed scale). This study adds to the state of the science for post-wildfire sediment forecasting as well as determining areas for sediment mitigation projects. A relatively novel sediment mitigation design (an alluvial fan restoration) is provided as an example of how these sediment forecasts can be successfully used to reduce impacts to both human and ecological values. The purpose of this study is of interest to an international audience and is similar to many other papers that validate (or invalidate) a particular model or theory for forecasting or predicting an environmental phenomenon.

This submission includes a track changed version of the manuscript incorporating all reviewer comments as well as a "clean" version of the manuscript.

**Reviewer One, second round of comments (initial authors' response in August 2024)**

**Authors' response in bold type the reviewer comment is in regular type, this response build on the previous round of reviews.**

**Comments on Revision**
The authors have addressed some of the comments from the last revision, but several major concerns remain with the existing manuscript. First, some of the literature citations are inappropriate. For example, they discuss sedimentation modeling and use a reference to a hydrology-only paper that does not address sedimentation directly (see line comments below). Second, the authors failed to address the comment about a timeline of events, and this remains important. This could be done relatively easily with a few additions to Figure 9. For example, in addition to showing the sediment, the authors could simply add a secondary axis with rainfall intensity that shows the timing of the major storms. Crucially, I think the authors need to also show the timing of the completion of their "alluvial fan

restoration areas" on Figure 9 as well. The third and final concern, is that the authors do not take the opportunity to explicitly compare observations with modeling. For example, everything in Table 1 appears to be a modeling result, but they have an opportunity to compare the model with observations.  Same comment with Figure 9. That shows sediment observations, but it'd make sense to compare these with the modeling. I realize that the modeling and observations are obtained over diKerent timescales, but you can sum the observations to the annual time scale and compare that to the modeling. Same comment with Table 3. Why not add a column that says "Sediment Retention Observed at Channel Design".

**The literature citations have been reviewed and revised accordingly (see line comments). The timeline of events is addressed throughout the paper, Figure 9 now shows flood events with the empirical sediment observations, Table 1 now includes the sediment transport results as well as the observed sediment, Section 2.1 now includes a brief discussion of the overall timeline of flooding, empirical observations, modeling, and mitigation structure development. While this improves the case study aspect of this paper it paradoxically is opposed to both reviewers' comments about improving this paper to provide a more clear scientific benefit. The purpose of this paper has been, and continues to be, to evaluate the efficacy of different sediment models for forecasting sediment risk and designing sediment mitigation strategies (as stated in the abstract and at the end of the introduction section). We believe the paper will help both applied flood management as well as the state of the science by demonstrating the utility and precision of various modeling techniques. Hopefully this will also help with refining these models (especially the easy-to-use WEPPcloud) to be more precise and more used internationally. Focusing on the case study timeline, or other details of the case study, distracts from the purpose of the paper. The purpose of presenting the mechanics of the case study is to provide enough methods to allow for the tested hypotheses to be replicated if so desired, other details beyond this objective are tangential to the paper.**

**We strongly disagree with the reviewer's third concern that the authors do not explicitly compare observations with modeling. Here are some example areas where the comparisons are made:**

**From the abstract: "***Empirical evidence from four floods in 2021 indicated 9,900 Mg of sediment yield to city of Flagstaff neighborhoods, WEPPcloud estimated 3870 Mg/year, MUSLE predicted 4860 Mg/year (based on the four events), and the WARSSS suite of models predicted 4630 Mg/year. Both WEPP and WARSSS estimated more sediment yield from channels than hillslope (51%/49% and 60%/40% respectively) though the spatial patterns differ between the models.***"**

**From the Results: "***The commonly used WEPP model demonstrated much lower sediment yields (3870 Mg/year) than the WARSSS model (4630 Mg/year) and empirical results (9900 Mg/year in 2021) for the Museum Fire burn scar and Spruce Wash watershed and slightly less than the event based MUSLE model (4860 Mg/year).***"**

**From the Discussion: "***All three modeling domains, MUSLE, WEPP, and WARSSS showed drastic increases in channel and hillslope sediment yields post-wildfire in this case study. Both WEPP and WARSSS predict slightly more sediment yield from existing channels than from hillslope processes, however the hillslope gullying and rill erosion is substantial. The similarity between model results, and less than an order of magnitude comparison with empirical results, indicate that both WEPPcloud and WARSSS are useful for sediment predictions.***"**

**From the Conclusions: "***This case study shows the utility of both WEPPcloud and WARSSS for predicting sediment transport to the city of Flagstaff, Arizona. The agreement between both models for sediment transport, and within an order of magnitude comparison to empirical observations from flood events in 2021, is encouraging. The difference between models was largely in the spatial pattern of sediment yield. Both models indicated a slightly higher contribution from channels than hillslopes but WARSSS, because it is partly empirically based, was better at identifying "hot spots" of both channel and hillslope sediment yield.***"**

**The results section provides the quantitative comparisons, the discussion section provides a commentary on the results, and the abstract and conclusions both summarize those comparisons (as is the format with nearly every peer-reviewed scientific paper). The comparison of the field observations with the modeled results is the focus of this paper, it is with some dismay that we read that the reviewer does not agree that enough to time is spent in this paper providing explicit comparisons of the modeling results to the empirical results. We have attempted to make the Introduction and Discussion even more explicit in the comparisons in this round of review but the paper is already written fairly straightforward.**

Right now my biggest concern is that this paper is a site-specific case study that does not have large application outside of the specific scenario. Moreover, I'm concerned that because the modeling is not compared to observations directly it's hard to evaluate how well these models actually estimated what was observed. Finally, I do not understand how the mitigation structures influenced/did not influence the sediment retention because I do not know when they were built with respect to the storms.

**As shown above there are ample comparison between the modeled results and the field observations. This paper uses a case study to demonstrate the utility and precision of three modeling techniques, that is the purpose of the paper (as explicitly stated in the Introduction) and has international applications as all three models can be used globally. Additionally, the WEPPcloud interface is free and extremely user friendly, any study that provides precision and accuracy results for this model could be very helpful to the larger flood risk community due to the lower barrier to entry on this model. The paper has been further improved (see line comments) to make the utility of the results even more explicit. Finally, in regards to the mitigation structures, this is a secondary purpose of the paper and a minor portion of the paper. The purpose of including the mitigation structures in the paper is to show the utility of the modeling techniques in designing and selecting locations for mitigation structures. We provided more detail on the "alluvial fan restoration" strategy as an introduction to the technique, since this case study used a relatively novel mitigation strategy. Line comments requested more references to this**

technique (mostly in regard to the Rebecca Beers conference paper reference) which we have attempted to improve upon. Since this technique is relatively novel there are no peer-reviewed articles that go into depth on the alluvial fan restoration technique and we are forced to use conference abstracts and technical reports. Again, the exact mitigation structure design is a minor component of this paper.

**Line comments**

29 I understand what you mean by sediment forecast here, but I don't understand what you mean by sediment forecast on line 28, where you say that "Sediment mitigation structures … are discussed as real-world applications of sediment forecasting…". I think this might just be a problem of language imprecision. I think you mean something closer to: "We discuss the real-world implications of using models to make sediment forecasts and using modeling results for the design of sediment mitigation structures." Either way, please refine this sentence because right now it sounds like you are saying that sediment mitigation structures are provide sediment forecasts, which doesn't make much sense.

**This sentence has been revised to provide clarity.**

34 Provide references to support this assertion, especially references that indicate that it is an "increasingly important issue", or change the language if that isn't something that is supported by research.

**The rise of catastrophic wildfire, especially at the WUI is very well documented. To prevent adding to the already long reference list we re-used existing references that document the rise in impacts as examples (specifically Ebel et al. 2023 which addresses it in the first sentence of their abstract and Sankey et al. 2017 which address it in the first two paragraphs of their introduction section).**

36. Sankey et al., 2017 is not a paper about flooding, and therefore you should adjust the reference. The word flood is not even used in that paper.

**The incorrect reference was used here, updated to Sankey et al. 2024 and an additional reference (Kinoshita et al. 2016) is added.**

40. Ebel et al., 2023 (which doesn't have a year in your references) is about hydrology and does not specifically address these concerns: "damaging debris flows and sediment sourcing, transport, and aggradation". Here and elsewhere, please use references that support the assertions you are making.

**The Ebel et al. 2023 paper discusses sediment sourcing modeling by calling it "erosion". Multiple models are discussed including WEPP, but we understand that one would have to read this paper comprehensively to parse out the sediment models from the hydrologic models. Ebel has been removed at this line location and two other more**

**clearly defined sediment model references have been included (Moody et al. 2013 and Smith et al. 2011).**

46. Same problem as the last two. These references aren't really well suited for the points you are trying to support.

**These three references were provided as examples of the large body of knowledge on sediment impacts on watershed ecosystem recovery.**

51. Either I'm confused about what you are referencing with Ebel et al., 2023, or you are confused. But I think you are trying to refer to this paper that is specifically focused on hydrological modeling and does not touch on sediment modeling at all. Is there a chance you are actually referring to a diKerent paper?:

Ebel, B. A., Shephard, Z. M., Walvoord, M. A., Murphy, S. F., Partridge, T. F., & Perkins, K. S. (2023). Modeling Post-Wildfire Hydrologic Response: Review and Future Directions for Applications of Physically Based Distributed Simulation. *Earth's Future*, *11*(2), e2022EF003038.

**The three instances of the Ebel reference in this paragraph have been removed, The Ebel paper does discuss sediment source modeling through their "erosion" discussion but we understand that this may be nuanced and have removed the reference to prevent confusion.**

62: Here you reference a conference abstract by Beers et al. 2023, and there is no mention of a loss in stream power or accretion upstream of neighborhoods in that conference abstract. Also, you inadvertently changed the title of that conference abstract, so I suggest you change it back to the original title. In any case, I don't think that the information in that abstract can be used to support the assertion here.

**We removed the Beers reference and added two other references (Grover 2021; Rosgen and Rosgen 2015). Since this mitigation technique is relatively novel we are forced to use conference proceedings and technical papers for any prior reference to the technique.**

103 When you say that the flood events allowed for empirical comparisons to the modeled predictions be more specific. Do you mean you compared sediment discharge, volume, mass, flood velocity, etc? Specify to readers what exactly could be compared.

**Added verbiage to explain that we are comparing sediment discharge.**

119: a 10-100 time increase in surface water runoK compared to what? Mean annual flow? Also, by runoK, do you mean discharge? Or depth? Be specific here.

**The wording has been changed to "peak discharge", there is no mean annual flow in these ephemeral channels as many years there was no flow at all. This only changed post-**

**wildfire, and even then flow duration is still measured in minutes and hours.**

126 There is something I don't understand about this: "Areas downstream from high sediment yield areas were identified as "work areas" …". I thought that the "work areas were specific mititation areas that were defined. This makes it sound like any area downstream from any other area with a high sediment yield is a "work area". I think you need to clarify this, and locate these areas on a map.

**This sentence provides the method of determining "work area" site selection. We added slightly more clarification (low gradient areas…). These areas are shown in Figure 6 in the Results section, a more appropriate section as the Methods should not be showing results.**

148: International readers of this European journal are likely not concerned about "who" estimated discharge, but I think they will be concerned about "how" it was estimated. So please include that information.

**The sentence has been revised to remove the name of the "who" and provide a little more context to the method. The full 2-D modeling method is quite long and complex. The references within the sentence at this line number provide the needed detail for recreating the discharge estimates.**

152: Are you saying that the floods incised the channels and then they will widen over time. Or are you saying that they were already incised prior to the fire, and will widen due to flooding. Please clarify.

**While the mechanics are not important (either of the interpretations of the sentence will provide the divergence in sediment models explained above this line item) we have clarified the sentence to explain that the channel incises post-wildfire and then widens into the future.**

154: here and elsewhere, when you are describing methods you have used, please use past-tense. For example, you say: "Sediment transport estimates are …", but that activity is over, so you should say "Sediment transport estimates were …". Same comment on 165 "sediment transport analyses are…" should be "sediment transport analyses were…"

**We corrected the tense in Line 154 but not at Line 165. At Line 165 the statement that sediment and flow data is difficult to collect for ungauged ephemeral streams is still relevant now and into the future. It is not a past-tense one time statement.**

207: Here you need to say the version number of the software, and provide a reference in your works cited.

**Added.**

214 say why you chose 1 inch, 2 inch and 3 inches per hour. Also, use "in" rather than a quote to abbreviate inches, as this will be more clear for an international audience.

**Rewritten to be more inclusive (better convey "inches"), the rationale for using standardized design storms is rather apparent and further explanation is not needed to explain this study. The manuscript is already fairly dense on the details of the methodology. Methods is meant to provide the reader the ability to recreate the study and to understand any weakness in the study design. Explaining the rationale for using every detail of a model is beyond what is needed to provide the study design to the reader.**

223 I don't think you can use these words together " measured qualitatively" because if it's qualitative it's not actually a measurement. So change to something like "estimated qualitatively". Also, specify the diKerence between sediment and debris here. Is debris wood and trash? Say how it's diKerent than sediment, or just remove the term debris.

**The sentence has been refined.**

234 after "…average annual sediment transport…" add units (e.g., Mg/yr) so readers know the units you are using.

**Added.**

243-244 I don't think you can justify this choice. In alluvial rivers the two-year return interval is often substituted for Bankfull, not the one year RI. Also what do you mean by "post-wildfire channel forming discharge" I don't think that's a concept that's been shown. What would be diKerent about a postfire channel forming discharge versus a non-postfire channel forming discharge? Why is "channel forming" relevant here? If you think these choices have merit, then I think you need to explain them or point to literature that explains them.

**This comment doesn't make sense, this paragraph is outlining how the model (in this case FLOWSED/POWERSED) was set up for this study. We don't have to explain the rationale for the individual model inputs, just what the inputs are in case a reader wants to re-create the experiment. Model parameters will be different for other watersheds around the world. Also, this case study is not on an alluvial river but on ephemeral mountain drainages that are rapidly adjusting to a watershed regime change in terms of soil, vegetation, and channel morphology character, a "typical" 2-year recurrence period for bankfull is not appropriate in these highly unstable high gradient systems.**

257: You need to provide this information in a table somewhere and then reference it here: "bankfull cross- sectional area, Manning's n value, bankfull discharge, slope, suspended sediment (mg/L), measured bankfull bedload (lb/s), a flow duration curve, and a sediment rating curve comparison"

**Since this data includes rating curves, sediment assemblages (ranges), and discrete site data it would be difficult to represent in a simplified summary table. A reference to the technical report is now provided instead. The Appendix (B) of this technical report includes 79 pages of supporting data that "feeds" the FLOWSED/POWERSED sediment**

**model. Similar to our comment above these model input values are only useful for recreating this case study and are not integral to understanding the precision and accuracy of the various models being tested in this study.**

263 I'm a little lost on how you would use a "bankfull flow" if you are trying to estimate flow on an alluvial fan. Please explain.

**This is explained in these two paragraphs in the manuscript but a little extra clarification has been added. The bankfull flow is used in the existing condition to calculate a baseline sediment load for the "work area" project. The "work area" (or alluvial fan restoration) is then modeled as a proposed condition to determine the benefit, in terms of reduced sediment flux. Bankfull flow is not determined for the proposed condition, the same flow rate is used as the existing condition (which is determined by bankfull calculations).**

285: Insteady of saying "G" channel, which is not a universally understood metric. I suggest you use a few short descriptions of the channel type. Something like "the channels are riKle-pool channels defined as "G" in XYZ scheme".

**Some clarification has been added for both "G" and "D" classifications. Both are "classic" incised (G type) or braided sediment laden low gradient (D type) channels.**

306 Avoid contractions like "don't" in scientific writing

**Corrected**

367 Earlier you said there were four events. Please correct or explain the discrepancy. On line 24 it says "MUSLE predicted 4860 Mg/year (based on the four events)"
**It was four events, this sentence has been corrected.**

374 Again the reasoning for these choices of rain events were not justified. Please state why you chose those particular storm amounts/durations.

**These were the actual storm events in 2021.**

398 State if field observations confirm the modeling here. I think that readers will want to know if the modeling matches what was observed at those cross sections.

**The alluvial fan restorations/"work areas" were not established until after the 2021 flow events. There are two timelines here: the modeling and verification through field observations, and the modeling to inform mitigation measures. This will be clarified in the Figure 9 timeline as recommended at the beginning of this review as well is in Section 2.1.**

408 I think the critical question here is how do the modeling results compare to field

observations. You have an opportunity here to compare model predictions with field observations, and that's the critical piece that seems to be missing.

**The modeling results compared to the field observations is provided multiple times (Abstract, Results, Discussion, and Conclusion). Unfortunately, we do not have field observations for the sediment transport/mitigation structure side of this study as the mitigation structures were built after the 2021 events. This is explained in this section ("Flow events in 2022 were muted in Spruce Wash due to small rain events, the alluvial fan sites that were constructed prior to monsoon season did appear to function well in terms of sediment aggradation and attenuation (Figure 10) however there were no flow events that over-topped the channel within the city to provide empirical comparisons.")**

**This is now furthered clarified in the Figure 9 which will provide a clear timeline of flood events and also in Section 2.1 which describes the flood events and the modeling and mitigation construction activities.**

416 This mention of the Pipeline needs more detail. How was the 70-80% measured? Was it repeat lidar, photogrammetry? Also, I think a critical piece is the timing after fire. Retaining 70-80% of sediment several years after the fire doesn't really say much about how the sediment retention structures would behave immediately after the fire. It is well known that wildfire sedimentation is typically highest in year 1 and goes down drastically in the following years as the watershed recovers.

**Since the authors did not conduct the field observations we cannot comment other than it was "repeat surveys" and sediment haul off (as referenced in the paper from Tiffany Construction LLC and the Coconino County Flood Control District). The Pipeline Fire impacted nine (9) watersheds so it is likely that multiple methods were employed in their estimates. We added timing considerations, however we must caveat that substantial sediment is observed after fires in this part of the world. Regular flooding and high sediment loads are still observed from other fires in the area of this case study. This is largely due to the vegetation state change in many of these areas (change from ponderosa pine to grassland). An in-depth discussion of post-wildfire watershed state change is outside the scope of this paper, but it is an interesting topic.**

450 What "hydrologic forecasts" are you referring to here?

**These are presented in the next two sentences with four references to the papers/reports that provided their predictions for future water yield in the project area.**

501 I don't think that prior research supports this assertion that there will be "…substantial sediment loading for the foreseeable future…" it is well known that sedimentation rates after fire decline precipitously after the first year after wildfire. Within 3 years it is very unlikely that you'll continue to have fire-related sedimentation problems. Provide evidence that would support this point if you want to make this assertion.

**The immediate post-fire sedimentation may decrease however these channels are now incised, the hillslopes now contain substantial gullies and rills, and the vegetation has transitioned from dense pine forest to sparse grassland. Studies in the region show**

**elevated sediment and flood risk for decades after a fire not to mention geomorphic adjustments. Geomorphic references are provided and we provided two new references that indicate uncertainty in long term sediment loads (McGuire review) and a comparison of flow events (which can function as a surrogate for sediment transport, from the 1977 Radio Fire and this current fire: JE Fuller 2024 technical report).**

507 Here and elsewhere, suggest replacing "poor conditions" with something else (e.g., transient or non-equilibrium conditions). I think you are trying to say that there is erosion or change, but calling it "poor" implies a judgement call on what is good/bad, which is really dependent on the observer.

**Replaced three instances of "poor" with "non-equilibrium".**

514 Again provide evidence of how you measured 70% sediment retention on the pipeline fire. Also, I just looked up the Pipeline fire and I can see that it burned starting in June 2022. Did you have alluvial fans built on that fire to capture sediment in 2022? If so state the date of when those sediment retention structures were built with respect to the date of the fire and storms.

**As mentioned in the manuscript this was not studied or observed by the authors, this was an observation of our County (regional government) counterparts at the nearby Pipeline Fire. There is a large body of applied research on the larger Pipeline Fire, including the efficacy of the alluvial fans, but to our knowledge none of that data has been published in a peer reviewed outlet at this time.**

526 Again, the Beers et al., 2023 reference is just a conference abstract so there is no additional detail to be found on this.

**See comment immediately above, the presentation is available upon request from the Arizona Geological Survey (Rebecca Beers). To clarify the data source we have added "personal comms. Rebecca Beers". Unfortunately, none of this data is currently published in a peer-reviewed journal (this current manuscript is for a fire three years previous than the Pipeline Fire and we are still in peer review). As an aside, one of our motivations for publishing this work is to provide techniques and data to the scientific community for both practical application as well as building the body of knowledge on post-wildfire sediment mitigation. Traditionally much of this work is completed by engineers and applied scientists who have little financial or professional incentive to publish their work (and in some cases publishing can be detrimental to their careers if the results are not to the liking of the funding agencies). There are probably a dozen papers worth of data, lessons learned, and techniques from recent fires in the Arizona, USA area alone, little of it will be preserved through publication unfortunately.**

527 Quantify what you mean by "moderate" and "long-term average"? Do you mean you retain 50% of the sediment or 98% and is this 50% of the long-term average measured using some dating technique or something else?

**We have revised this statement to demonstrate that these were field observations and not quantified. The authors have observed these structures working in multiple post-wildfire environments including the 2010 Schultz Fire, this fire (2019 Museum Fire), and the 2022 Pipeline Fire. Averaging the efficacy of the structures over multiple years we qualitatively observe that the structures retain a considerable portion of the upstream sediment load but that much fine sediment still passes through. Further investigation is obviously warranted but is not the aim or scope of this study (we are focusing on the model efficacy in this paper and how the models can inform mitigation design). The purpose of this sentence is to explain that we have observed these structures working and then transition into the next sentence caveating that high gradient alluvial fans might require maintenance during high intensity floods.**

Figure 4. Somewhere, maybe in a supplement. Provide definitions for all of these channel types.

**The caption already references the "Rosgen classification system" which is readily available, for free, on the internet through numerous venues. A general description of the channel type is also included in the caption to provide the reader with enough context to understand the watershed level pattern. Adding a supplement to this paper would be unnecessary.**

Figure 9. Please add rainfall intensity as a secondary axis on this so readers can compare the sediment mass per day with the rainfall intensity. You mention the four storms, but it'd be helpful to see where those storms exist in time compared to the sediment yield. Please also indicate the date that the alluvial fan restoration areas were completely constructed. You can do this with a single vertical line on the completion date.

**We reworked Figure 9 to show the peak flood flow at the downstream end of the modeling domain and the beginning of the built environment (the city neighborhoods). This provides the best comparison with the empirical observations of sediment. Rainfall is provided elsewhere in the manuscript and has been simplified into an average over the burned watershed (there are four rain gauges within the burned footprint). Adding rainfall as a third axis would make the figure overly "busy" while not providing any new information.**

**The alluvial fan "work areas", or restoration structure, were designed and built based on the models and were not built until 2022 and early 2023. The timeline will be explained better in the introduction or methods to show the difference between the sediment forecasting (conducted in 2021 and 2022), the empirical observations (conducted in 2021), and the construction of the mitigation structures (conducted in 2022). Please see Section 2.1 in the revised paper.**

**Reviewer 2, response to second round of comments (initial response on August 2024)**

**Authors' response in bold typeface.**

The authors have addressed some of the comments raised by the other reviewer and myself. Although this new version is clearly an improved piece of work, I think that this manuscript still needs some work before it can be accepted.

Overall, the context of this research is still too narrow. I would recommend the authors to zoom out from their (nice) case study and think of a broader audience. My comments on the abstracts is an example of such a broadening. The key point in an abstract is not necessary to provide details on model's name, but instead to say that several physically-based models have been run and compared to asses... Idem for the sediment yield values. What interest a reader is also to know if the values are normal, high; expected, similar to other cases? (etc.) Without such a broader perspective, the international dimenion of this research is missed in some way.

**The reviewer is missing the purpose of this study. As mentioned in the August response this study is intended to assess the precision and accuracy of commonly used sediment models using field observations from actual flood events to provide a measure of both precision and accuracy. The purpose is not to provide a measure of sediment load to compare to other post-wildfire environments. As mentioned previously the sediment load will vary based on numerous environmental factors (watershed size, burn severity, climate, soil, etc) that would make this study rather inconsequential if our purpose was only adding one more data point at a global level. Paradoxically the reviewer is requesting that this study become "merely" a case study by focusing on comparing sediment load with other case studies instead of comparing modeling techniques using actual flood events (a much more interesting study and more applicable globally). There are plenty of examples of published papers that validate, or invalidate, model predictions based on field measurements, this paper is written in the spirit of that line of scientific inquiry.**

**The authors of this paper have collectively worked dozens of post-wildfire assessments in the western USA and frequently have policy makers, land managers, scientists, and hydrologists ask if a certain sediment model is providing "correct" results or if the post-wildfire mitigation strategy (or strategies) are based on accurate modeling. The value in this study is showing that both WARSSS and WEPP are in general agreement between each other and within an order of magnitude of observed events. The WEPP model, in particular, is of interest to many as the WEPPcloud interface is extremely easy to use and has seen a great increase in use globally. Having a study that describes the precision and accuracy of this low-barrier-to-entry modeling regime is helpful not only for researchers in developed nations but also for applied science and management in areas with limited financial and engineering resources.**

Overall, I believe that many of my first round comments (and also those of the other reviewer) remain valid.

Smaller comments to clarify some of my requests/comments:
The study area maps would benefit form an information about the topography. I mentioned adding elevation quotes on the maps to have a broad ideas of the differences of elevation between the

summits and the valley floors.

**This was addressed in the August 2024 review, elevation is provided in the body of the text for some key locations. Adding elevations to the maps would make the maps difficult to read and also would provide nothing useful. The scale of the maps is large enough that any elevation label would necessarily span hundreds of meters of both vertical relief and horizontal distance, making the label useless. Modelers who will be reading this paper will be using similar digital elevational datasets as are referenced in the Methods section. The repeatability of this study hinges on those public datasets and not on elevational labels on relatively small maps within figures. Latitude and longitude is noted on each of the maps for quick reference in whatever digital elevation model (DEM) that a reviewer or reader would like to use for evaluating this case study. Paradoxically this repeated request for elevation data is contrary to the request that this study become less of a case study and more applicable globally.**

**August 2024 comments to a similar request by the reviewer:**

**Unsure of where the reviewer would like elevations called out, the topography is highly heterogenous. Example elevations for Mount Elden, Dry Lake Hills, Mount Elden Estates neighborhood, and Paradise/Sunnyside neighborhoods are now included in the Study Site section. Contour lines, and DEMs, for the area are freely available online at multiple sources (e.g. USGS National Map, Google Earth, City and County GIS portals, etc).**

In the discussion, gully erosion is mentioned. My comment on that was that a process that is discussed but not even mentioned on an earlier stage is something that is awkward.

**This was addressed in the August 2024 review, gully erosion is a component of hillslope erosion, a thesis on gully erosion is not the topic of this paper. Verbatim from the August 2024 review:**

**The hillslope gully erosion mentioned on this line is for the formation of hillslope gullies and rills through hillslope erosion. We are not talking about mass movement or mass wasting. No mention of landslides or debris flows is provided in the manuscript in terms of modeling or empirical observations. There are plenty of examples of gully erosion definitions in various government reports from agencies on multiple continents, this manuscript subscribes to the common definition of the term and is discussed at length in the Discussions section. Mentioning gully erosion as part of the hillslope erosion modeling is described in the Methods section.**
**The onset of gully erosion is interesting in this region due to the prevalence of gullying on hillslopes post-wildfire. The process is more fully explored in an earlier paper from the nearby Schultz Fire that is cited already in this manuscript. The reference for that paper is as follows:**

**Neary, D.G., Koestner, K.A., Youberg, A. and Koestner, P.E.: Post-fire rill and gully formation, Schultz Fire 2010, Arizona, USA. Geoderma, 191, pp.97-104, 2012.**

**Another reference to the Neary et al. study has been added at this line location.**